# AVERITEC: A Dataset for Real-world Claim Verification with Evidence from the Web

**Michael Schlichtkrull**\*, **Zhijiang Guo**\*, **Andreas Vlachos**
Department of Computer Science and Technology, University of Cambridge
{mss84,zg283,av308}@cam.ac.uk

## Abstract

Existing datasets for automated fact-checking have substantial limitations, such as relying on artificial claims, lacking annotations for evidence and intermediate reasoning, or including evidence published after the claim. In this paper, we introduce AVERITEC, a new dataset of 4,568 real-world claims covering fact-checks by 50 different organizations. Each claim is annotated with question-answer pairs supported by evidence available online, as well as textual justifications explaining how the evidence combines to produce a verdict. Through a multi-round annotation process, we avoid common pitfalls including context dependence, evidence insufficiency, and temporal leakage, and reach a substantial inter-annotator agreement of $\kappa = 0.619$ on verdicts. We develop a baseline as well as an evaluation scheme for verifying claims through question-answering against the open web.

## 1 Introduction

Fact-checking is considered crucial for limiting the impact of misinformation [Lewandowsky et al., 2020]. Unfortunately, not enough resources are available for manual fact-checking. Automated fact-checking (AFC) has been proposed as an assistive tool for fact-checkers, moderators, and citizen journalists to facilitate it [Cohen et al., 2011, Vlachos and Riedel, 2014], inspiring applications in journalism [Miranda et al., 2019, Dudfield, 2020, Nakov et al., 2021] and other domains, e.g. science [Wadden et al., 2020].

Substantial progress has been made on common benchmarks, such as FEVER [Thorne et al., 2018] and MultiFC [Augenstein et al., 2019]. Nevertheless, existing resources have recently come under criticism. Many datasets (for example, Thorne et al. [2018], Schuster et al. [2021], Aly et al. [2021]) contain purpose-made claims derived from sources such as Wikipedia, and are thus unlike real-world claims checked by journalists. Further, in these datasets, *refuted* claims are produced by corrupting existing sentences. Datasets that do contain real-world claims either lack evidence annotation [Wang, 2017], or annotate it superficially using automated means, resulting in issues such as including evidence published days or weeks *after* the investigated claims [Glockner et al., 2022].

To address these limitations we introduce AVERITEC (Automated VERIfication of TExtual Claims), which combines real-world claims with realistic evidence retrieved from the web, as well as justifications for veracity labels. We formulate retrieval as question generation and answering, providing a structured representation of the evidence and reasoning supporting or refuting the claim. The free-text justifications detail how the evidence is used to reach the verdict, including cases of conflicting evidence, matching best practises for human fact-checkers [Borel, 2016]. In constructing AVERITEC, we ameliorate three issues afflicting existing datasets with real-world claims:

1. **Context Dependence:** Ousidhoum et al. [2022] found that claims in some datasets based on fact-checking articles (e.g., Fan et al. [2020]) cannot be verified without additional

---

\*Equal Contribution.

37th Conference on Neural Information Processing Systems (NeurIPS 2023) Track on Datasets and Benchmarks.

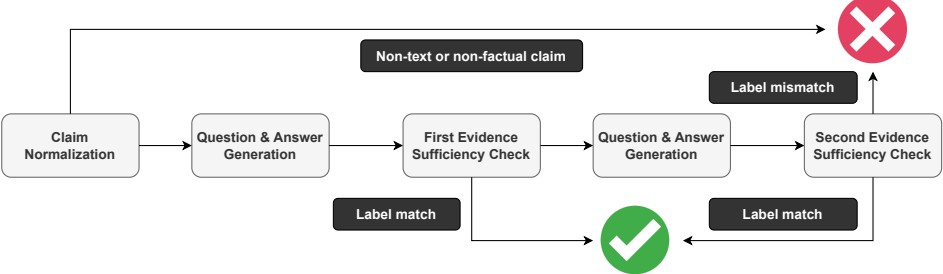

Figure 1: Diagram for our annotation process. Claims are first selected and normalized. Then, two rounds of question-answer pair generation and evidence sufficiency check ensure high-quality evidence annotation.

    information from the articles they were extracted from. This could for example be due to unresolved coreference or ellipsis, e.g. in "unemployment is rising" it is unclear which geographical/temporal context is being considered.

2. **Evidence Insufficiency:** Glockner et al. [2022] found that labels in some datasets (e.g., Hanselowski et al. [2019]) often do not match the annotated evidence, because they rely on e.g., assumptions about the speaker of the claim. This is significantly different from *not enough evidence* verdicts, which is a label for claims where evidence could not be found.

3. **Temporal Leaks:** Glockner et al. [2022] found that annotations in some datasets (e.g., Augenstein et al. [2019]) contain leaks from the future. For example, a claim from January might be annotated with evidence from March that year. Leaks can also happen between splits, if e.g., evidence for a training claim from 2021 also pertains to a test claim from 2020.

We address (1) through an initial normalisation step, where annotators enrich claims with necessary information from the fact-checking article. We verify the adequacy of this, and address (2), by combining multiple rounds of annotation with a "blind" quality control step, re-annotating any claims with insufficient evidence. We address (3) by restricting annotators to evidence documents published before the claim, and by ordering our training, development, and test splits temporally. With the increasing reliance on large pretrained language models, temporal ordering provides an additional benefit: if the training data for the language model is cut off before the temporal start of the test set, leaks from pretraining cannot occur either.

AVERITEC consists of 4,568 examples, collected from 50 fact-checking organizations using the Google FactCheck Claim Search API[2]; itself based on ClaimReview[3]. Our annotation, which involved up to five annotators per claim, resulted in substantial inter-annotator agreement, with a free-marginal $\kappa$ of 0.619 [Randolph, 2005]. We further develop a baseline to explore the feasibility of the task, relying on Google Search, BM25 [Robertson and Zaragoza, 2009], retrieved in-context prompts [Liu et al., 2022, Rubin et al., 2022], and a trained stance detection model. AVERITEC is the first AFC dataset to provide both question-answer decomposition and justifications, as well as avoid issues of context dependence, evidence insufficiency, and temporal leaks. Our dataset and baseline are available under a CC-BY-NC-4.0 license at `https://github.com/MichSchli/AVeriTeC`.

## 2 Related Work

Sourcing real-world claims from fact-checking articles is popular (e.g. Wang [2017]), as extracting claims from fact-checkers guarantees *checkworthiness*. That is, any claim included in the resulting dataset is deemed interesting enough to be worth the time of a professional journalist (see Hassan et al. [2015]). Previous real-world datasets either lack annotations for evidence, or suffer from context dependence, evidence insufficiency, or temporal leaks. Further, they do not provide annotations for intermediate steps, and only a minority (Alhindi et al. [2018], Kotonya and Toni [2020]) provide justifications. A comparison between AVeriTeC and prior datasets can be seen in Table 1.

[2]`https://toolbox.google.com/factcheck/apis`, available under a CC-BY-4.0 license.
[3]`https://www.claimreviewproject.com/`

| Dataset | Claim | | | Evidence | | |
| --- | --- | --- | --- | --- | --- | --- |
| | *Source* | *Type* | *Independence* | *Sufficient* | *Unleaked* | *Retrieved* |
| FEVER [Thorne et al., 2018] | Wikipedia | Synthetic | ✓ | ✓ | N/A | ✓ |
| VitaminC [Schuster et al., 2021] | Wikipedia | Synthetic | ✓ | ✓ | N/A | ✓ |
| FEVEROUS [Aly et al., 2021] | Wikipedia | Synthetic | ✓ | ✓ | N/A | ✓ |
| SciFact [Wadden et al., 2020] | Science | Synthetic | ✓ | ✓ | N/A | ✓ |
| FM2 [Saakyan et al., 2021] | Game | Synthetic | ✓ | ✓ | N/A | ✓ |
| Covid-Fact [Saakyan et al., 2021] | Reddit | Synthetic | ✓ | ✓ | N/A | ✓ |
| Liar-Plus [Alhindi et al., 2018] | Factcheck | Real | ✗ | ✓ | ✗ | ✗ |
| PolitiHop [Ostrowski et al., 2021] | Factcheck | Real | ✗ | ✓ | ✗ | ✗ |
| MultiFC [Augenstein et al., 2019] | Factcheck | Real | ✗ | ✗ | ✗ | ✓ |
| XFact [Gupta and Srikumar, 2021] | Factcheck | Real | ✗ | ✗ | ✗ | ✓ |
| PubHealth [Kotonya and Toni, 2020] | Factcheck | Real | ✗ | ✗ | ✓ | ✗ |
| WatClaimCheck [Khan et al., 2022] | Factcheck | Real | ✗ | ✗ | ✓ | ✗ |
| ClaimDecomp [Chen et al., 2022] | Factcheck | Real | ✗ | ✗ | ✓ | ✗ |
| Snopes [Hanselowski et al., 2019] | Factcheck | Real | ✗ | ✗ | ✓ | ✗ |
| QABrief [Fan et al., 2020] | Factcheck | Real | ✗ | ✗ | ✓ | ✗ |
| ClimateFEVER [Diggelmann et al., 2020] | Web | Real | ✗ | ✗ | ✓ | ✓ |
| HealthVer [Sarrouti et al., 2021] | Web | Real | ✗ | ✗ | ✓ | ✓ |
| CHEF [Hu et al., 2022] | Factcheck | Real | ✗ | ✓ | ✗ | ✓ |
| AVERITEC | Factcheck | Real | ✓ | ✓ | ✓ | ✓ |

Table 1: Comparison of fact-checking datasets. *Source* indicates where the claims are collected from, such as Wikipedia, or fact-checking articles (Factcheck). *Type* indicates whether the claims are synthetic or real-world. *Independence* indicates whether the claim is context independent. *Sufficient* indicates whether the evidence can provide sufficient information. *Unleaked* means whether the evidence contains leaks from the future and *retrieved* denotes whether the dataset involves evidence retrieval instead of relying on pre-retrieved passages e.g. the fact-checking article.

Beyond evidence insufficiency and temporal leakage, Glockner et al. [2022] also found that many examples require a *source guarantee* to refute, i.e. a guarantee that the claimant's underlying reason for making the claim is known to the debunker. For example, evidence against the claim *"COVID-19 vaccines may kill sharks"* can only be found when incorporating the underlying reasoning of the claimant, that the manufacturing of COVID-19 vaccines requires a chemical extracted from sharks. We do not explicitly provide such a guarantee; however, as each claim in AVERITEC is annotated with the original claimant, these underlying reasons can be recovered through question-answer pairs.

Question-answer decomposition is considered a promising strategy; Yang et al. [2022] proposed such a model even without a dataset of annotated question-answer pairs. Two recent datasets cast fact-checking as question-answering: Fan et al. [2020] and Chen et al. [2022]. However, Fan et al.'s [2020] question-answer pairs were only written to add relevant context, not to capture entire the fact-checking process, and thus lack evidence sufficiency. Ousidhoum et al. [2022] furthermore identified context dependence as a significant concern in Fan et al. [2020]: many questions are impossible to generate given only the claim, as they refer to entities and events only mentioned in the original fact-checking article. Chen et al. [2022] did – like us – attempt to ensure evidence sufficiency. However, they take no steps to verify their success. Furthermore, their evidence is taken directly from the fact-checking articles which are written after the claim, thus exhibiting temporal leakage.

## 3   Annotation Structure

Our dataset consists of 4,568 real-world claims annotated with question-answer pairs representing the evidence, a veracity label, and a textual justification describing how the evidence supports the label. An example can be seen in Figure 2.

Reasoning about evidence is represented through questions and answers. Questions may have multiple answers, a natural way to show potential disagreements in the evidence. Questions can refer to previous questions, allowing for multi-hop reasoning. Answers (other than *"No answer could be found."*) must be supported by a *source url* linking to a web document. To avoid sources disappearing from the web, we cache all pages used as evidence in the internet archive[4].

Claims in AFC datasets are typically *supported* or *refuted* by evidence, or there is *not enough evidence*. We add a fourth class: *conflicting evidence/cherry-picking*. This covers both conflicting evidence, and

---

[4]https://archive.org

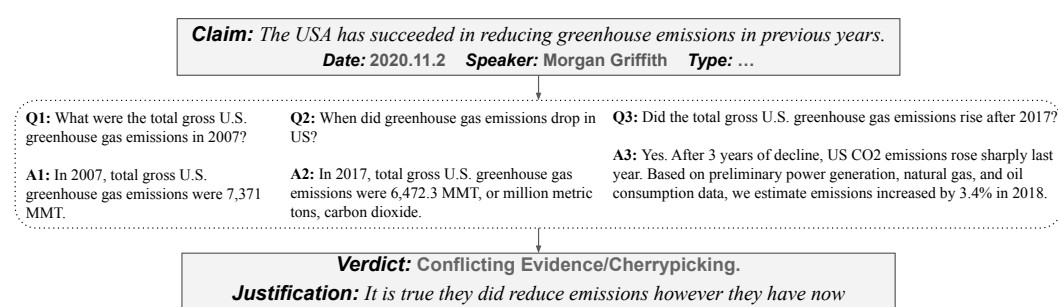

Figure 2: Example claim from AVERITEC. As opposed to previous datasets, ours naturally combines question-answer pairs that break down the evidence retrieval with justifications that show how evidence leads to verdicts.

technically true claims that mislead by excluding important context. For real-world claims, sources may interpret events differently, and therefore legitimately disagree. This differs from Schuster et al. [2021], which studies claims for which the evidence has been revised. Adding a fourth class has also recently been discussed in the context of natural language inference [Jiang and Marneffe, 2022], although there it is usually ambiguity in the premise or hypothesis leading to the conflict.

AVERITEC also provides textual justifications that explain how verdicts are reached from the evidence. Where sources disagree, best practices for established fact-checkers is to provide a textual explanation of *why* the claim misleads [Uscinski and Butler, 2013, Amazeen, 2015]. These justifications can be *substantial* [Toulmin, 1958], i.e. they may introduce logical leaps supported by commonsense or inductive reasoning beyond the retrieved evidence. For example, if a claim states that 50% of a population group were vaccinated by February 1st, and evidence shows only 33% had been vaccinated by January 31st, the justification may reason that a 50% rate one day later is unlikely.

We include several fields of metadata: the *speaker* of the claim, the *publisher* of the claim, the *date* the claim was published, and the *location* most relevant to the claim. These can be used to support questions, answers, and justifications. We also annotate the *claim type* and *fact-checking strategy* of each claim. Type represents common aspects, e.g., whether claims are about numerical facts; strategy represents the approach of the fact-checkers, e.g., whether they relied on expert testimony. Types and strategies should not be used as input to models (at inference time), but can provide useful data for analysis.

## 4 Annotation Process

Starting from 8,000 fact-checking articles, we first identified and discarded 537 duplicates and 802 paywalled or dead articles. We passed the remainder through a five-phase pipeline – see Figure 1. First, an annotator extracts claims and relevant metadata from each article, providing context independence. Second, an annotator generates questions and answers them using the web. These annotators also choose a temporary verdict. Third, a different annotator provides a justification and a verdict based *solely* on the annotated question-answer pairs; this serves as an evidence sufficiency check. Any claim for which the two verdicts do not match is passed through the last two phases again. If the verdicts still disagree, the claim is discarded. Different annotators were used for each claim in each phase – i.e., no annotator saw the same claim twice. Our annotation was performed by the company Appen[5]; details and annotation guidelines can be found in Appendices C and J.

**Claim Extraction & Normalisation**  Annotators extracted the central claims from each fact-checking article, enriching them with the necessary context. This is necessary as many fact-checking articles cover multiple claims (e.g., several rumours circulating about an event). Further, some claims lack adequate contextualization [Ousidhoum et al., 2022]. For example, the claim *"we have 21 million unemployed"* requires coreference resolution. After claim extraction, we discarded *speculative* claims,

---

[5]https://appen.com/

| Split | Train | Dev | Test |
|---|---|---|---|
| Claims | 3068 | 500 | 1000 |
| Questions / Claim | 2.60 | 2.57 | 2.57 |
| Reannotated (%) | 28.1 | 24.4 | 25.1 |
| End date | 25-08-2020 | 31-10-2020 | 22-12-2021 |
| Labels (S / R / C / N) (%) | 27.6 / 56.8 / 6.4 / 9.2 | 24.4 / 61.0 / 7.6 / 7.0 | 25.5 / 62.0 / 6.3 / 6.2 |

Table 2: Descriptive statistics for the dataset. Statistics for labels are split into supported (S), refuted (R), conflicting evidence/cherrypicking (C), and not enough evidence (N). For the dev and test splits, the start date is the end date of the previous split; the train set has no start date.

i.e. unverifiable statements about future events or personal opinions, and *multimodal* claims, i.e. claims where the type or the strategy inherently involves modalities beyond text.

**Question Generation & Answering**   Annotators generate questions and answer them providing evidence about the claim. The aim is to "deconstruct" the reasoning of the fact-checker into a QA-structure, extracting question-answer pairs that match the content of their enquiries. This makes the process better amenable to annotators who are not trained journalists, and provides structured representations for model development and evaluation. Each question must be accompanied by an answer (or marked *unanswerable*), and answers must be backed by sources. We advise annotators that extractive answers are preferred, but abstractive answers are also allowed. Annotators in this phase are asked to provide a verdict based on their retrieved evidence, possibly different from the one in the fact-checking article.

As annotators follow the fact-checking articles, the ideal evidence sources for answer are the documents linked from the articles. For answers are not found in the linked documents, we provide access to a custom Google search bar. This search bar is restricted to show documents published only *before* the claim date, in contrast to prior work [Fan et al., 2020]. Furthermore, unlike Alhindi et al. [2018] and Hanselowski et al. [2019], the fact-checking article itself cannot be used as evidence.

We note that, where annotators could not find the publication date of the *claim*, their instructions were to use the publication date of the *fact-checking article* instead. As the process of fact-checking can take journalists several days, there is a window in which news about the claim can be published that we cannot prevent from being used as evidence. Further, the Google API does not always correctly infer the publication dates of articles. As such, our guarantee against temporal leakage is approximate (see Section 8). For the development set, we estimated that around 6% of answers are sourced from a fact-checking domain. This is primarily because of the earlier publication of an article about the same claim by a different fact-checking organization.

**Evidence Sufficiency Check**   Once question-answer pairs have been generated, we present each claim along with its question-answer pairs to a new annotator. This annotator does *not* see the fact-checking article. They then produce a verdict and a textual justification for it. We compare this verdict to the one produced by the question and answer annotator, and if they disagree we repeat the question-&-answer generation and sufficiency check steps with new annotators to improve the evidence and the verdict.

## 5   Dataset Statistics

We split our dataset into training, validation, and test data temporally (see Table 2). Claims have on average 2.60 questions, and questions have on average 1.07 answers. Most answers (53%) are extractive, followed by abstractive (26%) and boolean (17%) answers. A few questions (4%) are marked unanswerable as no available evidence could be found by the annotators. Statistics for source document modality, fact-checker strategy, and claim type can be seen in Appendix F. We note that AVERITEC is somewhat unbalanced – the majority of claims are *refuted*. This is a consequence of our choice to rely on fact-checking articles, as journalists tend to pick false or misleading claims to work on. Our dataset includes all ClaimReview claims labeled *supported* (or any variation thereof, e.g. *true*) within our temporal limits (for more detail see Appendix I.1).

To measure the inter-annotator agreement of our annotation scheme, we had a second set of annotators re-annotate 100 claims from the dataset. In this we assumed the claim extraction and normalization step was done, and the annotators repeated the question and answer generation phase. Since we have an unbalanced dataset, following Kazemi et al. [2021], Ousidhoum et al. [2022], we therefore measure agreement using Randolph's [2005] free-marginal multirater $\kappa$, an alternative to Fleiss' $\kappa$ more suitable for unbalanced datasets [Warrens, 2010]. Our observed agreement score of $\kappa = 0.619$ is substantial, and compares well to those for other "hard" fact-checking annotation tasks, e.g. Kazemi et al. [2021], who got between $\kappa = .30$ and $\kappa = .63$ depending on the language. Using Fleiss' $\kappa$ [Fleiss, 1971], we get an agreement score of $0.503$.

# 6   Evaluation

To evaluate models on AVERITEC, we follow Thorne et al. [2018] and score retrieved evidence based on agreement with gold evidence, and give credit to veracity predictions (and justifications) only when correct evidence has been found. However, unlike in FEVER and other datasets using a closed source of evidence such as Wikipedia, AVERITEC is intended for use with evidence retrieved from the open web. Since the same evidence may be found in different sources, we cannot rely on exact matching to score retrieved evidence. As such, we instead rely on approximate matching.

To measure how well a set of generated questions and answers match the references, we rely on a pairwise scoring function $f : S \times S \to \mathbb{R}$, where $S$ is the set of sequences of tokens. We then use the Hungarian Algorithm [Kuhn, 1955] to find an optimal matching of generated sequences to reference sequences. Formally, let $X : \hat{Y} \times Y \to \{0, 1\}$ be a boolean function denoting the assignment between the generated sequences $\hat{Y}$ and the reference sequences $Y$. Then, the total score $u$ is calculated as:

$$u_f(\hat{Y}, Y) = \frac{1}{|Y|} \max \sum_{\hat{y} \in \hat{Y}} \sum_{y \in Y} f(\hat{y}, y) X(\hat{y}, y) \tag{1}$$

If $f$ is an exact match, we recover the evidence *recall* score from Thorne et al. [2018]. Our metric is as such a generalization of theirs to the approximate case. In our evaluation, we use the implementation of METEOR [Banerjee and Lavie, 2005] in NLTK [Bird et al., 2009] as the scoring function $f$ (and refer to our evidence scoring function as Hungarian METEOR hereafter), but any suitable pairwise metric could be used. We chose METEOR over other alternatives (e.g., ROUGE [Lin, 2004]) as it is known to correlate well with human judgments of similarity [Fomicheva and Specia, 2019]. We do not employ a precision metric, as we want to avoid penalizing systems for asking additional relevant information-seeking questions – however, all systems are limited to a maximum of $k = 10$ question-answer pairs.

We conduct the evaluation with Hungarian METEOR twice: once using only the questions as input sequences, and once using the concatenation of questions and answers. A subtask of AVERITEC is to *ask the right questions* – as we discuss in Section 7.2, good questions are very useful as search queries even if not accompanied by a good answer. Finding the right angle to criticize a claim is a substantial task by itself; it covers the creativity factor in retrieval discussed by Arnold [2020]. Including the question-only score allows comparison of systems along this axis as well. To evaluate veracity predictions and justifications, we use a cutoff of $f(\hat{y}, y) \geq \lambda$ to determine whether correct evidence has been retrieved (using concatenated questions and answers); any claim for which the evidence score is lower receives veracity and justification scores of $0$.

Many claims can be verified through alternative evidence formulations. Taking an example from the 100 claims annotated twice for Section 5, one annotator might produce the question-answer pair *"Where did South Africa rank in alcohol consumption? In 2016, South Africa ranked ninth out of 53 African countries."* while another produces *"What's the average alcohol consumption per person in South Africa? 7.1 litres."*. These may both be valid ways of establishing the relative levels of alcohol consumption between South Africa and other countries. We recognize that our evaluation approach can penalize systems for selecting an alternative evidence path; nevertheless, we argue that automatic evaluation on this task is helpful in model development. We note that a similar phenomenon was seen for the original FEVER dataset [Thorne et al., 2018], despite the artificial claims and the exclusive use of Wikipedia as an evidence source. There, the authors suggested crowd-sourced human evaluation as a more reliable alternative – we echo their recommendation. Our annotation process hints at a potential setup for human evaluation: judging if a body of evidence is sufficient for a verdict is exactly what our annotators did during the evidence sufficiency check phase.

| Model | Q only | Q + A | Veracity @ (.2/.25/.3) | | | Justifications @ (.2/.25/.3) | | |
|---|---|---|---|---|---|---|---|---|
| No search | 0.19 | 0.11 | 0.03 | 0.02 | 0.01 | 0.02 | 0.01 | 0.01 |
| Gold evidence | 1.00 | 1.00 | 0.49 | 0.49 | 0.49 | 0.28 | 0.28 | 0.28 |
| AVERITEC -BLOOM-7b | 0.26 | 0.21 | 0.23 | 0.15 | 0.00 | 0.11 | 0.07 | 0.05 |
| gpt-3.5-turbo | 0.29 | 0.16 | 0.17 | 0.10 | 0.06 | 0.06 | 0.04 | 0.02 |

Table 3: Results for the AVERITEC baseline and ChatGPT (gpt3.5-turbo). Retrieval scores both for questions and for questions + answers are given in terms of Hungarian METEOR score. Veracity and justifications are scored using accuracy and METEOR respectively, in both cases conditioned on correct evidence retrieved at $\lambda = \{0.2, 0.25, 0.3\}$ (see Section 6). We report results for three versions of the baseline, as discussed in Section 7.2: a version that uses no evidence (no search), a version that uses gold evidence (gold evidence), and the full pipeline described in Section 7.1 (AVERITEC). We also report results for gpt-3.5-turbo (ChatGPT).

We further note that our metric is straightforward to extend to cover evaluation with multiple reference sets. Given a set of sets of question-answer pairs $R$ representing different questioning strategies, a best-matching score could be computed as $\max_{Y \in R} u_f(\hat{Y}, Y)$. As such, if AVERITEC was expanded with annotations for alternative questioning strategies, our metric could score models on these as well.

To understand how our metric should be interpreted, we also computed Hungarian METEOR scores between the question-answer pairs generated during the two rounds of annotation used for inter-annotator agreement in Section 5. At $0.28$ for questions and $0.22$ for questions and answers, these results are quite low, highlighting the difficulty of automatic evaluation for this task. Investigating claims with low agreement scores, we see that these are actually often a result of different annotators using different evidence sources for the same verdict, or phrasing equivalent question-answer pairs differently. Based on our observations of human annotations, we recommend $\lambda = 0.25$ as an appropriate cutoff value for our metric. We refer to this metric (for veracity prediction) as AVERITEC score.

# 7 Experiments

## 7.1 Baseline Model

Our baseline is a pipeline consisting of several components: generation of search questions, search, generation of questions given retrieved evidence, reranking of retrieved evidence, veracity prediction, and generation of justifications. For each step in the pipeline, we carried out experiments with several models. Using our training set, we finetuned, respectively, BERT-large [Devlin et al., 2019] for classification (340M parameters) and BART-large [Lewis et al., 2020] for generation (406M parameters). We furthermore tried a few-shot setup, prompting a large language model (LLM) with retrieved in-context examples [Liu et al., 2022, Rubin et al., 2022] from our training set. Here, we tried the 7b parameter BLOOM model [Scao et al., 2022] and the 13b parameter Vicuna model [Chiang et al., 2023]. We limited ourselves to relatively small models, as we consider runnability crucial for a baseline: all our components can be run on a single Nvidia A100 GPU. As such, our baseline strikes a balance between performance and computational cost.

**Search** Given a claim, we retrieve evidence documents from the internet using the Google Search API. Following Karadzhov et al. [2017], we use a reduced version of the claim keeping only verbs, nouns, and adjectives as the search term. As we did during annotation, we limit the API to documents published *before* the estimated date of the claim. We keep all unique documents in the first 30 search results. Initial experiments showed that questions were very useful as additional search terms. As the model does not have access to gold questions during testing, we instead generate questions. We experimented with three models: BART-large, *bloom-7b*, and *Vicuna-13b*. Surprisingly, BLOOM performed the best, beating the newer and larger Vicuna; we attribute this primarily to greater topical diversity in the set of questions generated, and thus greater variety in the retrieved evidence pages. We rely on prompting with retrieved in-context examples [Liu et al., 2022, Rubin et al., 2022]. We use BM25 [Robertson and Zaragoza, 2009] to find the 10 most similar claims from the training set,

and use their annotated questions to construct a prompt, with which we generate questions for the claim using BLOOM. We tested {1,3,5,10} in-context examples, finding 10 to perform the best. The full prompt can be seen in Appendix D.1. We add any new unique documents retrieved by searching for these generated questions. This can be seen as a form of query expansion [Mao et al., 2021].

**Evidence Selection**    Once a set of evidence documents has been created for each claim, we pick $N = 3$ sentences from this set. We first apply a coarse filter to the evidence set, ranking evidence sentences by BM25 score computed against the claim, and discard those outside the top 100. Then, we generate a question for each sentence that is answerable *by* that sentence, again using BLOOM. We tested {1,3,5,10} in-context examples, finding 10 to perform the best. The full prompt can be seen in Appendix D.2. We then re-rank these question-answer pairs to find the ones most relevant for the claim, using a finetuned BERT-large model [Devlin et al., 2019] (for more details, see Appendix E.1). This somewhat counter-intuitive strategy of retrieving first and then generating questions can be seen as using the generated questions to bridge claims to distantly related evidence sentences. Our approach is similar to the document expansion strategy proposed for question answering in Nogueira et al. [2019], except applied for reranking rather than the initial retrieval step.

**Veracity Prediction**    Once question-answer pairs have been generated, we produce verdicts through a stance detection strategy inspired by past work on filtering evidence [Hanselowski et al., 2019]. We use a finetuned BERT-large model to label each question-answer pair as supporting, refuting, or being irrelevant to the question (for more details, see Appendix E.2). We then deterministically label the claim as follows: 1) if the claim has both supporting and refuting evidence, label it *conflicting evidence/cherrypicking*. 2) If the claim has only supporting question-answer pairs, label it *supported*; similar for *refuted*. 3) Otherwise, label the claim *not enough evidence*. We tested three different models for veracity prediction: BERT-large, *bloom-7b1*, and *Vicuna-13b. We found BERT to perform better by a slight margin; using gold evidence, we obtained macro-F1 scores of .49, .43, and .48 for the three models respectively.*

**Justification Generation**    The final step is to generate a textual justification for the verdict. Here, we rely on BART-large [Lewis et al., 2020] finetuned on our training set (for more details, see Appendix E.3). We use the concatenation of the claim and the retrieved evidence as input; we tried adding the predicted veracity as well, but saw no improvements to performance. We again tested three models: BART-large, *bloom-7b1*, and *Vicuna-13b*, respectively obtaining a METEOR score of .28, .23, and .25. Based on our qualitative analysis of 20 claims, the justifications generated by Vicuna are very good, but the model is penalized for being overly verbose – Vicuna generates 36 tokens on average, compared to 21 in the gold data.

## 7.2    Results

We evaluate as discussed in Section 6. We include results in Table 3 at three thresholds for comparison, although we encourage $\lambda = 0.25$. We compare our baseline to two other models: one without access to search, and one using gold question-answer pairs as evidence.

For the *no search* model, we use prompting to generate questions, following the approach described in Appendix D.1. We leave all answers as *"No answer could be found"*. Generating answers is not an option, as answers must be supported by sources. We use BERT-large finetuned on the training data to predict veracity labels (without any evidence), and the same prompting strategy as discussed in Section 7.1 to generate justifications. For the *gold evidence* model, we use the gold question-answer pairs provided by our annotators in the place of generated questions and retrieved evidence. That is, we test only the veracity prediction and justification production components.

Analysing retrieval results on the development set, we still find $\lambda = 0.25$ to be a good cutoff point for the AVERITEC veracity and justification metrics. For borderline question-answer pairs this threshold is high enough that all important information must be produced to meet it, but there is still some room for paraphrasing and partial evidence.

Our baseline has decent performance at $\lambda = 0.2$ and $\lambda = 0.25$, but does not perform well at higher evidence cutoff points. Because of the structure of our pipeline – generate search terms, retrieve and rerank evidence, generate questions to match the reranked evidence – our baseline struggles to match specific evidence sets. If the retrieved evidence paragraph is very short, e.g., a table cell reading

| Model | S | R | C | N | Macro |
|---|---|---|---|---|---|
| No evidence | .30 | .22 | .00 | .16 | .17 |
| Gold evidence | .48 | .74 | .15 | .59 | .49 |
| AVERITEC | .41 | .69 | .10 | .16 | .23 |
| gpt-3.5-turbo | .62 | .71 | .02 | .20 | .39 |

Table 4: *F1*-scores for veracity prediction split across labels: supported (S), refuted (R), conflicting evidence/cherrypicking (C), and not enough evidence (N). We also show the macro-average. Again, we report results for three versions of the baseline (see Section 7.2): a version that uses no evidence (no search), a version that uses gold evidence (gold evidence), and the full pipeline ( AVERITEC ). We also report results for gpt-3.5-turbo (ChatGPT).

"January 24th", the question generation model often lacks context to generate the right question. Further, the baseline cannot generate questions with highly abstractive answers, only questions that can be answered directly by sentences in the supporting sources.

We recognize that the retrieval scores of this baseline are quite close to those of the human annotators seen in Section 6. Nevertheless, for evidence retrieved by our baseline, low scores are much more frequently a result of reliance on *wrong* evidence, rather than *equivalent* evidence phrased differently and scored incorrectly. Further research is needed to develop an evaluation capable of recognizing this difference, e.g., a trained metric in the style of BLEURT [Sellam et al., 2020].

The gap between gold evidence and retrieved evidence highlights how retrieval remains challenging, also discussed in Arnold [2020]. Manually analysing 20 examples from the development set, we find that Google search results based on the claim and the generated questions contain useful evidence only in 9/20 cases. If the retrieval system had access to the gold questions for use as search queries, correct evidence would be found in 16/20 of these cases; this highlights the need for further development of retrieval and search systems, and especially query/question generation.

We also report individual *F1* scores for each veracity class (as well as a macro average) in Table 4. Our veracity prediction model fails to accurately predict *Conflicting Evidence/Cherrypicking* most of the time, even with gold evidence. Going through the predictions, we see that precision is very low (10% using gold evidence). Labelling claims as *Conflicting Evidence/Cherrypicking* if any evidence is classified as having different stance leads to many false positives – often, questions that simply add context to supported claims are (incorrectly) labelled as *refuting* by the stance detection model.

As a way to improve the stance detection component, we tried to generate additional training data using gpt-3.5-turbo (ChatGPT). We paraphrased each claim in the dataset, using the same evidence. We generated one paraphrase per claim. Then, we trained BERT-large on the concatenated original and paraphrased claims. Unfortunately, this failed to yield additional performance, producing a macro-*F1* score of .46 on gold data; slightly lower than the .49 obtained using only the original claims. The primary cause is a drop in performance for *refuted* and *not enough evidence*, which the model trained on paraphrased data conflates more often.

We further include results in Tables 3 and 4 using ChatGPT. As ChatGPT cannot produce sources to back up its answers, this is not directly comparable to our baseline. Nevertheless, it is an interesting point of comparison. We generate evidence and verdicts with ChatGPT, using the prompt described in Appendix G. We find that ChatGPT outperforms our baseline in terms of pure question generation, but nevertheless received a lower AVERITEC score (veracity prediction at $\lambda = 0.25$). This is a consequence of the missing retriever: generated answers often do not match gold answers (i.e., they are either alternative correct answers, or outright hallucinations).

ChatGPT performs well on veracity, especially for supported claims; but those verdicts are often not supported by valid evidence. For example, for the claim *"1 cup of dandelion greens = 535% of your daily recommended vitamin K and 112% of vitamin A."*, ChatGPT assigned the verdict *supported* and generated the evidence string *"According to the USDA, 1 cup of chopped dandelion greens provides 535% of your daily recommended vitamin K and 338% of vitamin A, which is higher than the claim"*. While the verdict is true, there is no such statement from the USDA, and the actual gold evidence relies on several question-answer pairs and a calculation to arrive at the verdict.

# 8 Limitations

The evaluation metric we have presented alongside AVERITEC contains a significant limitation: no efforts are made to ensure answers and source documents are consistent. As only 53% of gold answers are fully extractive, it is expected that abstractive models will be employed. Such models though can hallucinate, and can thus make up answers that are not supported by the underlying sources, which our evaluation metric cannot detect. Further research is needed on evaluation to counteract this, along with research on developing an evaluation strategy that better allows verifying claims correctly with different questioning strategies and evidence documents.

While claims geographically concern regions from all around the world, all fact-checking sources and consequently all claims used in our dataset are in English. Further, as we take claims directly from fact-checking articles, our dataset is subject to any biases present within those articles; notably, for internal fact-checking, Barnoy and Reich [2019] documented a selection bias resulting from journalists rating claims by male sources more credible than female sources.

Finally, we note that our reliance on Google Search to avoid temporal leakage is a noisy process. The dates we rely on are the best estimate computed by Google[6]. As such, while in general evidence documents were available when associated claims were published, there may be exceptions.

# 9 Ethics Statement

Fact-checking is often envisioned as an epistemic tool, limiting the spread and influence of misinformation. The datasets and models described in this paper are not intended for truth-telling, e.g. for the design of automated content moderation systems. The labels and justifications included with this dataset relate only to the evidence recovered by annotators, and as such are subject to the biases of annotators and journalists; furthermore, the machine learning models and search engine used for the baseline contain well-known biases [Noble, 2018, Bender et al., 2021]. Acting on veracity estimates arrived at through biased means, including automatically produced ranking decisions for evidence retrieval, risks causing epistemic harm [Schlichtkrull et al., 2023].

Annotators for our dataset had access to searching the entire web when finding evidence documents. We curated a list of common misinformative sources by combining several public documents[7], and flagged search results from these sources. Nevertheless, we did not prevent annotators from using them as evidence. Pointing out that a claim originates from an untrustworthy site is an important fact-checking strategy, and, indeed, our list may well contain false positives. A total of 85 answers in AVERITEC rely on a flagged source; moreover, our list is not complete. Our dataset may as such include misleading examples, and can potentially cause harm if relied on as an authoritative source.

We did not take any steps to anonymise the data. The claims discussed in our dataset are based on publicly available data from journalistic publications, and concern public figures and events – references to these are important to fact-check claims. We did not contact these public figures, or the journalists who published the original fact-checking articles. If any person included in our dataset as a speaker of a claim, as the subject of a claim, or as the author of a fact-checking article a claim is based on requests it, we will remove that claim from the dataset.

# 10 Conclusion

We have introduced AVERITEC, a new real-world fact-checking dataset consisting of 4,568 claims, each annotated with question-answer pairs decomposing the fact-checking process, as well as justifications. Our multi-step annotation process guarantees high-quality annotations, providing evidence sufficiency and avoiding temporal leakage; it also results in a substantial inter-annotator agreement of $\kappa = 0.619$. We have also introduced and analysed a baseline as well as an evaluation scheme, establishing AVERITEC as a new benchmark.

---

[6]See `https://developers.google.com/search/blog/2019/03/help-google-search-know-best-date-for`

[7]We combined the following: `https://libguides.castleton.edu/evaluating_news/fake_news`, `https://www.factcheck.org/2017/07/websites-post-fake-satirical-stories/`, and `https://en.wikipedia.org/wiki/List_of_fake_news_websites`

## Acknowledgements

We would like to thank Zhangdie Yuan, Pietro Lesci, Nedjma Ousidhoum, Ieva Staliunaite, and David Corney for their helpful comments, discussions, and feedback. This research was supported by the ERC grant AVeriTeC (GA 865958). Andreas Vlachos is further supported by the EU H2020 grant MONITIO (GA 965576).

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

# A Dataset Access

We release our dataset and baseline at `https://github.com/MichSchli/AVeriTeC`, and will maintain it there. As we anticipate using the dataset in a future shared task, we are as of submission time only releasing the training and development splits. We will make the test split available privately to reviewers upon request.

# B Author Statement

The authors of this paper bear all responsibility in case of violation of copyrights associated with the AVERITECdataset.

# C Annotation Details

We carried out our annotations with the help of Appen (`https://appen.com/`), an Australian private company delivering machine learning products. The annotations took place on a special-purpose platform developed by our team and supplied to Appen. We will make the code for this platform available upon request. Appen provides guarantees that annotators are paid fairly: see `https://success.appen.com/hc/en-us/articles/9557008940941-Guide-to-Fair-Pay`. We spent a total of €40,835 for crowdworkers in our annotation process.

# D Baseline Prompts

## D.1 Claim Question Generation

To enrich our search results, we generate additional questions for use as search queries. For each claim, we retrieve the 10 most similar claims from the training dataset (computed using BM25). We combine these into a prompt following the scheme shown in Figure 3. We incorporate both the speaker and the claim itself in a form of preliminary experiments found to be highly effective: *"Outrageously, SPEAKER claimed that CLAIM. Criticism includes questions like: "*. The adversarial tone encourages the model to generate questions useful for debunking – we found this to be crucial for finding additional useful search results beyond those returned using the claim itself.

## D.2 Passage Question Generation

Once search results have been found, we generate questions for each line of each searched document using the process described in Section 7.1. We retrieve the 10 most similar question-answer pairs from the training dataset (computed using BM25 between the answer and the evidence line). We combine these into a prompt following the scheme shown in Figure 3. We experimented also with including the claim when generating the questions, however, we found this to decrease performance by acting as a distractor; BLOOM would generate questions related only to the claim and unrelated to the evidence. Passage question generation was by far the most expensive part of our experiments. While we made sure the model fits in memory of an A100 GPU, we parallelized inference across several. Using eight GPUs, question generation took approximately 24 hours.

## D.3 Justification Generation

We use a further prompt to generate justifications given the claim and verdict for the no-evidence baseline. Again, for each claim, we retrieve the 10 most similar claims from the training dataset (computed using BM25). We experimented with the same adversarial form discussed for question generation in Appendix D.1, but did not see any improvements in performance.

# E Baseline Models

We finetuned models for several components of our baseline. The following sections list hyper-parameter settings for each of those models. All training took place on a single Nvidia A100 GPU.

Outrageously, United We Win claimed that "As mayor, Cory Booker, US Democratic presidential candidate and US New Jersey Senator, passed criminal justice reform in a US Republican Senate.". Criticism includes questions like: Did U.S. Sen. Cory Booker pass criminal justice reform? What was the Republicans' majority in the Senate when the bill was signed into law?

Outrageously, Mokwele Kholofelo Banny claimed that "A married couple in Florida, Tito and Amanda Watts, was arrested for selling fake golden tickets to heaven to hundreds of people.". Criticism includes questions like: Did married couple Tito and Amanda Watts from Florida get arrested for selling fake golden tickets to heaven?

Outrageously, Muhammadu Buhari claimed that "It makes no sense for oil to be cheaper in Nigeria than in Saudi Arabia.". Criticism includes questions like: What was the price of petrol in Nigeria in Oct 2020? What was the price of petrol in Saudi Arabia in Oct 2020?

...

Outrageously, Tea talk and gossip claimed that "Microsoft cofounder Bill Gates said Be nice to nerds. Chances are you'll end up working for one.". Criticism includes questions like: Is Bill Gates quoted as saying "Be nice to nerds, chances are you'll end up working for one"?

Outrageously, Sen. Amy Klobuchar claimed that "US President Trump called for reduced funding for the Centre for Disease Control and Prevention.". Criticism includes questions like: Did US President Trump propose budget cuts in the funding for the Centre for Disease Control and Prevention?

Outrageously, US Democratic presidential candidate Wayne Messam claimed that "It is illegal for mayors to even bring up gun reform for discussion in Florida, US.". Criticism includes questions like:

Figure 3: Example prompt used to generate search questions for the claim *"It is illegal for mayors to even bring up gun reform for discussion in Florida, US."* with the speaker *"US Democratic presidential candidate Wayne Messam"*.

### E.1 Evidence Reranking

We used the BERT-large model [Devlin et al., 2019] with a text classification head, relying on the huggingface implementation [Wolf et al., 2020]. The model has 340 million parameters. We finetuned the model using Adam [Kingma and Ba, 2015] with a learning rate of $0.001$ and a batch size of $128$. The evidence reranker is trained using negative sampling. For each triple of claim $c$, question $q$, and answer $a$, we construct three negatives by corrupting each of $c$, $q$, or $a$, for a total of 9 negative samples per positive. Corrupted elements are replaced with randomly selected others from the dataset.

### E.2 Stance Detection

The setup for the stance detection model is similar to the evidence reranker. We again used the BERT-large model [Devlin et al., 2019] with a text classification head, relying on the huggingface implementation [Wolf et al., 2020]. The model has 340 million parameters. We finetuned the model using Adam [Kingma and Ba, 2015] with a learning rate of $0.001$ and a batch size of $128$. To train the stance detection model, we constructed examples from the training set. For claims with *supported* labels, we created one example per question for a positive stance. For claims with *refuted* labels, we created one example per question for negative stance. For claims with *not enough evidence* labels, we created one example per question for a neutral stance. Finally, we discarded all claims with *conflicting evidence/cherrypicking as the label*.

> Evidence: The image of Time magazine cover with Rachel Levine as woman of the year was posted on Facebook by "The United Spot", which is labelled as a satire site. Question answered: Which website said that Rachel Levine was Time's Woman of the Year?
>
> Evidence: Yes, because the wording was actually "complete 57 mega dams". Question answered: In 2017, did the Kenyan Government manifesto say they would construct 57 mega dams?
>
> Evidence: No, because the blog text uses future terminology like "...the bill is being brought in..." and "...this nz food bill will pave the way...". Question answered: Does the blog post imply that this Food Bill is already legislation?
>
> ...
>
> Evidence: China described the reports from Pakistan as "Baseless & fake". Question answered: Did China report any losses relating to this clash?
>
> Evidence: After carrying a few boxes that appeared full of supplies, Pence was informed that the rest of the boxes in the van were empty and that his task was complete. "Well, can I carry the empty ones? Just for the cameras?" Pence joked. "Absolutely," an aide said as the group laughed. Pence then shuts the doors to the van and returns to talk to facility members from the nursing home. Question answered: Were the PPE boxes that Mike Pence delivered empty?
>
> Evidence: Kris tells the magazine Caitlyn was "miserable" and "pissed off" during the last years of their marriage. Question answered:

Figure 4: Example prompt used to generate a question for the evidence line *"Kris tells the magazine Caitlyn was "miserable" and "pissed off" during the last years of their marriage."*.

### E.3 Justification Generation

For the justification generation model, we used the BART-large model [Lewis et al., 2020]. As previously we relied on the huggingface implementation [Wolf et al., 2020]. BART-large has 406M parameters. We finetuned the model using Adam [Kingma and Ba, 2015] with a learning rate of 0.001 and a batch size of 128. When generating, we used beam search with 2 beams and a maximum generation length of 100 tokens.

## F   Dataset statistics

To analyse our dataset, we computed various statistics for each dataset split. An overview of modalities in which evidence was found can be seen in Table 5. Statistics for claim type and fact-checker strategy can be found in Tables  and  respectively.

Annotators rely on evidence from a wide variety of different sources, taking evidence from a total of 2989 different domains. Interestingly, the most frequent is twitter.com (3%), typically representing announcements from public officials. This is followed by africacheck.org (2.5%), as Africa Check relies to a greater extent on references to its own past articles. After this follow official sources (e.g. cdc.gov (1.5%), who.int (1.3%), gov.uk (0.7%), wikipedia.org (1.4%)) and news media (e.g. nytimes.com (1.1%), washingtonpost.com (0.7%), and reuters.com (0.6%)). An interesting occurrence is a small number of non-textual sources, e.g. youtube.com (0.8%).

## G   ChatGPT Prompts

For the prompt used for our gpt-3.5-turbo experiments, see Figure 6.

Claim: A married couple in Florida, Tito and Amanda Watts, was arrested for selling fake
golden tickets to heaven to hundreds of people.
Our verdict: Refuted.
Our reasoning: The answer and source clearly explain that it was an April Fool's joke so the
claim is refuted.

Claim: North Korea blew up the office used for South Korea talks.
Our verdict: Supported.
Our reasoning: The building used was indeed destroyed.

...

Claim: US President Trump called for reduced funding for the Centre for Disease Control
and Prevention.
Our verdict: Supported.
Our reasoning: From the source, I saw tangible evidence where it stated that there was a
proposal by US President Trump to slash more than $1.2 billion of CDC's budget.

Claim: It is illegal for mayors to even bring up gun reform for discussion in Florida, US.
Our verdict: Conflicting Evidence/Cherrypicking.
Our reasoning:

Figure 5: Example prompt used to generate a justification for the claim *"It is illegal for mayors to
even bring up gun reform for discussion in Florida, US.".* Evidence and verdict for the claim are
produced in previous stages of the pipeline.

|  | Train | Dev | Test |
|---|---|---|---|
| Web text: | 68.2 | 75.5 | 74.9 |
| PDF: | 11.9 | 7.7 | 9.7 |
| Metadata: | 6.1 | 5.9 | 5.0 |
| Web table: | 4.9 | 3.0 | 2.9 |
| Video: | 1.1 | 1.1 | 1.9 |
| Image/graphic: | 2.0 | 2.7 | 1.6 |
| Audio: | 0.1 | 0.0 | 0.8 |
| Other: | 1.3 | 1.4 | 0.2 |
| Unanswerable: | 4.5 | 2.8 | 3.0 |

Table 5: Evidence modalities (%)

Can you fact-check a claim for me? Classify the given claim into four labels: "true", "false",
"not enough evidence" or "conflicting evidence/cherrypicking". Let's think step by step.
Provide justification before giving the label. Given claim:

It is illegal for mayors to even bring up gun reform for discussion in Florida, US.

Figure 6: Prompt used to generate evidence and verdicts with ChatGPT for the example claim *"It is
illegal for mayors to even bring up gun reform for discussion in Florida, US.".*

|                       | Train | Dev  | Test |
|-----------------------|-------|------|------|
| Position Statement    | 7.8   | 5.8  | 7.0  |
| Numerical Claim       | 33.7  | 23.8 | 21.8 |
| Event/Property Claim  | 57.8  | 61.4 | 69.8 |
| Quote Verification    | 9.6   | 13.8 | 7.7  |
| Causal Claim          | 11.5  | 10.8 | 11.9 |

Table 6: Claim types (%)

|                        | Train | Dev  | Test |
|------------------------|-------|------|------|
| Written Evidence       | 78.8  | 88.6 | 88.0 |
| Numerical Comparison   | 30.6  | 19.0 | 19.2 |
| Fact-checker Reference | 6.6   | 7.4  | 7.7  |
| Expert Consultation    | 29.9  | 27.4 | 29.6 |
| Satirical Source       | 3.6   | 2.0  | 1.8  |

Table 7: Fact-checker strategies (%)

# H   Additional Results

## H.1   Claim type

We computed baseline performance in terms of veracity at different evidence thresholds for each claim type. Results can be seen in Table 9 below:

# I   Data Statement

Following Bender and Friedman [2018], we include a data statement describing the characteristics of AVERITEC.

## I.1   Curation Rationale

We processed a total of 8,000 texts from the Google FactCheck Claim Search API, which collects English-language articles from fact-checking organizations around the world. We selected claims in the two-year interval between 1/1/2022 and 1/1/2020. Within that span, we selected all claims marked *true* by fact-checking organizations, as well as a random selection of other claims; this was done to reduce the label imbalance as much as possible.

We discarded claims in several rounds. First, any duplicate claims were discarded using string matching. Then, annotators discarded paywalled claims, as well as claims about or requiring evidence from modalities beyond text. Finally, we discarded any claim for which agreement on a label could not be found after two rounds of annotation.

## I.2   Language variety

We include data from 50 different fact-checking organizations around the world. While our data is exclusively English, the editing standards used at different publications differ. As such, several varieties of news domain English should be expected; given the distribution of fact-checkers involved, these will be dominated by *en-US*, *en-IN*, *en-GB*, and *en-ZA*.

## I.3   Speaker demographics

We did not analyse the demographics of the individual speakers for each claim. However, we asked annotators to specify the location most relevant to the claims. The distribution can be seen in Table 10.

| | Fraction of claims |
|---|---|
| africacheck.org: | 0.154 |
| politifact.com: | 0.153 |
| leadstories.com: | 0.096 |
| fullfact.org: | 0.068 |
| factcheck.afp.com: | 0.062 |
| factcheck.org: | 0.050 |
| checkyourfact.com: | 0.041 |
| misbar.com: | 0.032 |
| washingtonpost.com: | 0.029 |
| boomlive.in: | 0.026 |
| dubawa.org: | 0.023 |
| polygraph.info: | 0.020 |
| usatoday.com: | 0.019 |
| altnews.in: | 0.019 |
| indiatoday.in: | 0.019 |
| newsmeter.in: | 0.018 |
| newsmobile.in: | 0.015 |
| factly.in: | 0.015 |
| vishvasnews.com: | 0.015 |
| aap.com.au: | 0.014 |
| thelogicalindian.com: | 0.013 |
| verafiles.org: | 0.011 |
| nytimes.com: | 0.011 |
| healthfeedback.org: | 0.011 |
| thequint.com: | 0.008 |
| newsweek.com: | 0.005 |
| icirnigeria.org: | 0.005 |
| bbc.co.uk: | 0.004 |
| factcheck.thedispatch.com: | 0.004 |
| ghanafact.com: | 0.003 |
| factcheckni.org: | 0.003 |
| theferret.scot: | 0.003 |
| rappler.com: | 0.003 |
| covid19facts.ca: | 0.003 |
| newsmobile.in:80: | 0.002 |
| thegazette.com: | 0.002 |
| abc.net.au: | 0.002 |
| ha-asia.com: | 0.002 |
| sciencefeedback.co: | 0.001 |
| cbsnews.com: | 0.001 |
| fit.thequint.com: | 0.001 |
| namibiafactcheck.org.na: | 0.001 |
| thejournal.ie: | 0.001 |
| poynter.org: | 0.001 |
| zimfact.org: | 0.001 |
| climatefeedback.org: | 0.001 |
| factchecker.in: | 0.001 |
| pesacheck.org: | 0.001 |
| ghana.dubawa.org: | 0.001 |
| scroll.in: | 0.001 |

Table 8: Fact-checking sites used

| | $\lambda = 0.2$ | $\lambda = 0.3$ |
|---|---|---|
| Quote Verification | .13 | 0.7 |
| Numerical Claim | .17 | .10 |
| Event/Property Claim | .13 | .06 |
| Causal Claim | .11 | .04 |
| Position Statement | .10 | .04 |

Table 9: Baseline performance on each claim type, computed with two different evidence standards.

## I.4   Annotator demographics

For this dataset, we relied on the company *Appen* to provide annotators. Although the company itself is headquartered in Australia, demographic details regarding location or nationality of the annotators

| Country code | Count |
| --- | --- |
| US: | 1937 |
| IN: | 536 |
| GB: | 305 |
| KE: | 293 |
| NG: | 280 |
| ZA: | 191 |
| PH: | 73 |
| AU: | 56 |
| CN: | 55 |
| RU: | 38 |
| CA: | 31 |
| NZ: | 23 |
| GH: | 17 |
| IE: | 17 |
| LK: | 14 |
| TH: | 12 |
| FR: | 12 |
| PK: | 12 |
| IL: | 11 |
| IT: | 10 |
| DE: | 8 |
| ZW: | 7 |
| HK: | 7 |
| MM: | 6 |
| BR: | 6 |
| UA: | 6 |
| KR: | 5 |
| JP: | 5 |
| KP: | 5 |
| PL: | 5 |
| None: | 501 |

Table 10: Count of locations appearing in our dataset. All countries are listed using ISO country codes. Countries with fewer than five occurences are excluded – we will provide this data upon request.

were unfortunately not shared with us. We employed a total of 25 annotators with an average age of 42, and a gender split of 64% women and 36% men.

## I.5 Speech situation

The original claims were uttered in a variety of situations. We did not track this statistic for the entire dataset. However, analyzing a randomly selected 20 claims from our dataset, the majority (11) are social media posts. 4 originate from public speeches by politicians, 3 from newspaper articles, 1 from a political candidate's website, and 1 from a viral YouTube video.

The claims were all chosen by fact-checking organizations for analysis, and presented in a journalistic format on their websites.

## I.6 Text characteristics

We compute various statistics for the text included in this dataset; see Section 5 and Appendix F. The genre is a mix of political statements, social media posts, and news articles (see the previous subsection).

# J Annotation Guidelines

## J.1 Introduction

We aim to construct a dataset for automated fact-checking with the following guiding principles. First, we intend to decompose the evidence retrieval process into multiple steps, annotating each individual step as a question-answer pair (see Figure 2). Second, our dataset will be constructed from real-world claims previously checked by journalistic organisations, rather than the artificially created claims used in prior work.

Decomposing claim verification into generations and answering questions allows us to break complex real-world claims down to their components, simplifying the task. For example, in Figure 2, verifying the claim requires knowing the salary of the health commissioner, the governor, the vice president, and Dr. Fauci, so that they can be compared. Four separate questions about salary need to be asked in order to reach a verdict (i.e. that the claim is *supported*).

By decomposing the evidence retrieval process in this way, we also produce a natural way for systems to justify their verdicts and explain their reasoning to users. In addition to this, we annotate claims with a final justification, providing a textual explanation of how to combine the retrieved answers to reach a verdict. This allows users to follow each step of the retrieval and verification processes, and so understand the reasoning employed by the system.

---

*Claim:* **Biden lead disappears in** NV, AZ, GA, PA **on 11 November 2020.**

*Q1: Which media project Biden will win in Nevada?*
*A1:* ABC News, CBS News, NBC News, CNN, Fox News, Decision Desk HQ, Associated Press, Reuters, and New York Times.

*Q2: Which media project Biden will win in Arizona?*
*A2:* Fox News and Associated Pre.

*Q3: Which media project Biden will win in Georgia?*
*A3:* None.

*Q4: Which media project Biden will win in Pennsylvania?*
*A4:* ABC News, CBS News, NBC News, CNN, Fox News, Decision Desk HQ, Associated Press, Reuters, and New York Times.

*Verdict:* Refuted
*Justification:* Many media organizations believe Biden will win in NV, AZ, and PA. As such, his lead has not disappeared.

---

Figure 7: Example claim and question-answer pairs.

The annotation consists of the following three phases:

1. Claim Normalization.
2. Question Generation.
3. Quality Control.

Each claim should be annotated by different annotators in each phase. An annotator can participate *in* in all three phases, but they will be assigned different claims.

**Warning!** Components of the AVeriTeC annotation tool may not render correctly in some browsers, specifically *Opera Mini*. If this is an issue we recommend trying another browser, e.g. Firefox, Chrome, Safari, or regular Opera.

## J.2   Sign In

Each annotator will have received an **ID** and a **Password** with the access link to the annotation server. The password can be changed after logging into the interface.

**Important!**

- Make sure to log out at the end of the session!

- Do not open multiple tabs/windows of the AVeriTeC annotation tool. Always use only one window during annotation! If you are logged into multiple sessions using the same account, the annotation tool may lose the data you enter.

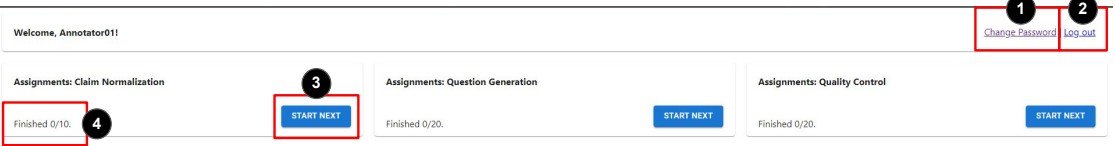

Figure 8: Interface of the control panel. (1) Button for changing the password. (2) Button for logout. (3) Start the annotation for this phase. Here is Phase 1 Claim Normalization. (4) The left number shows how many claims have been annotated and the right number shows how many claims are assigned for the current annotator at this phase.

After clicking the **START NEXT** button, the annotation phase will start. If an annotator is new to the current phase, the interface will provide a guided tour as in Figure 9 for that phase. Please read the hints provided by the tour guide carefully before the annotation.

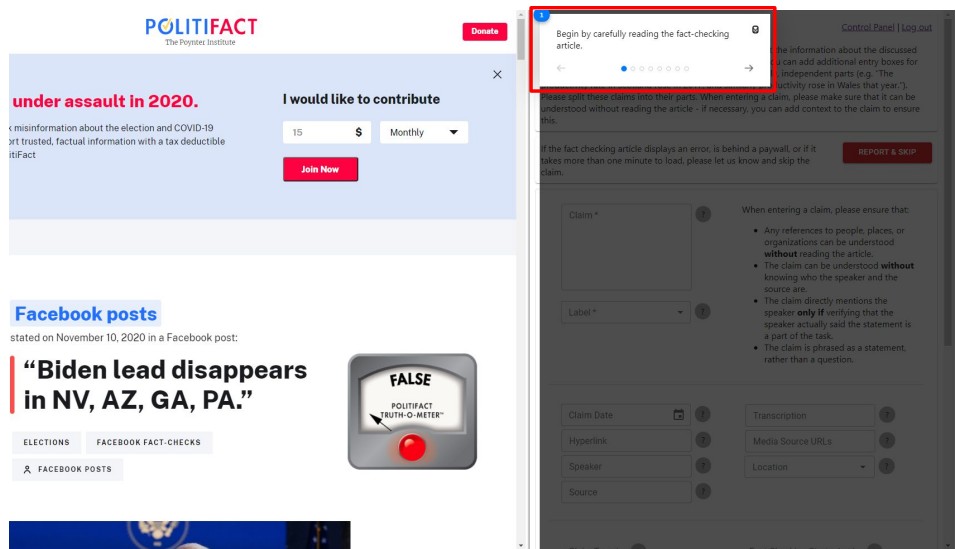

Figure 9: Interface of the tour guide.

### J.3 Phase 1: Claim Normalization

In the first phase, annotators collect metadata about the claims and produce a normalized version of each claim, as shown in Figure 10. The first step is to identify the claim(s) in the fact-checking article. Often, this can be found either in the headline or explicitly in some other place in the fact-checking article. In some cases, there may be a discrepancy between the article and the original claim (e.g. the original claim could be *"there are 30 days in March"*, while the fact-checking article might have the headline *"actually, there are 31 days in March"*). In those cases, it is important to use the *original* version of the claim. If there is ambiguity in the article over the exact wording of the claim, annotators should use their own judgment.

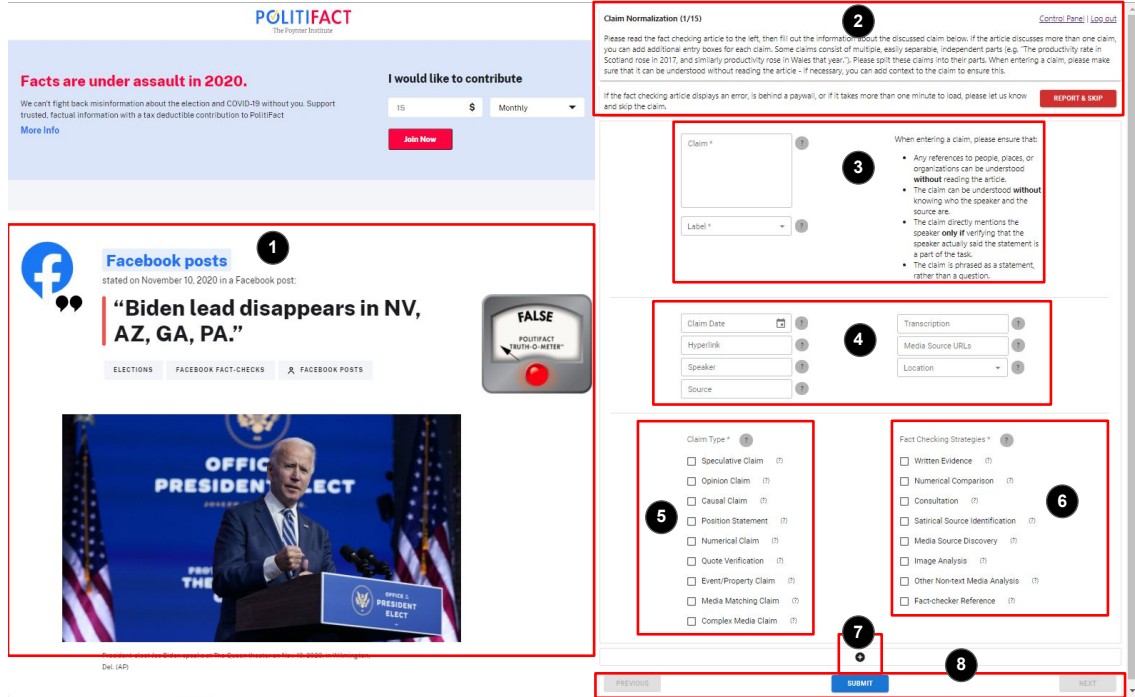

Figure 10: Interface of claim normalization. ①  The fact-checking article provided. ②  Guideline of annotation for this phase. Please read it before annotating. Notice that if the article displays a 404 page or another error, or if it takes more than one minute to load, please click the **REPORT & SKIP** button. ③  Fields for the normalized claim and the corresponding label. ④  General information of the claim. ⑤  Check-boxes for selecting the type of the claim. ⑥  Check-boxes for selecting the fact-checking strategy used. ⑦  Button for adding more claims. ⑧  Buttons for submitting the current claim, going to the previous claim, and the next claim.

#### J.3.1 Overview

Here, we give a quick overview of the claim normalization task; an in-detail discussion can be found in subsequent sections. Further documentation can also be found on-the-fly using the tooltips in the annotation interface.

1. First, annotators should read the fact-checking article and identify which claims are being investigated.

2. If the fact-checking article is paywalled or inaccessible due to a 404-page or a similar error message, annotators should report this and skip the claim using the provided button. We warn that some fact-checking articles can take too long to load – as such, while fact-checking articles that do not load at all should be skipped, we ask annotators to wait for at least one minute before skipping an article while it is still trying to load.

3. Most articles focus on one claim. However, some articles investigate multiple claims, or claims with multiple parts – in those cases, annotators should first split these into their parts (see Section J.3.2).

4. Some claims cannot be understood without the context of the fact-checking article, e.g. because they refer to entities not mentioned by name in the claim. In those cases, annotators should add context to the claims (see Section J.3.3).

5. Generally, we prefer claims to be as close as possible to their original form (i.e. the form originally said, *not* the form used in the fact-checking article). As such, contextualization should be done only when necessary, following the checklist in Section J.3.3.

6. Annotators should extract the verdict assigned to the claim in the article and translate it as closely as possible to one of our four labels – *supported*, *refuted*, *not enough evidence*, or *conflicting evidence/cherry picking* (see Section J.3.4). In phase one, annotators should give their own judgments – rather, they should match as closely as possible the judgments given by the fact-checking articles.

7. Claims will have associated metadata, i.e. the date the original claim was made, or the name of the person who made it. Annotators should identify and extract this metadata from the article (see Section J.3.6).

8. Annotators should identify the type of each claim, choosing from the options described in Section J.3.8. These are not mutually exclusive, and more than one claim type can be chosen.

9. Annotators should identify the strategies used in the fact-checking article to verify each claim, choosing from the options described in Section J.3.9. These are not mutually exclusive, and more than one claim type can be chosen.

### J.3.2 Claim Splitting

Some claims consist of multiple, easily separable, independent parts (e.g. *"The productivity rate in Scotland rose in 2017, and similarly productivity rose in Wales that year."*). The first step is to split these compound claims into individual claims. Metadata collection and normalization will then be done independently for each individual claim, and in subsequent phases, they will be treated as separate claims.

When splitting a claim, it is important to ensure that each part is understandable without requiring the others as context. This can be done either by adding metadata in the appropriate field, such as the claimed speaker or claim date, or through rewriting. For example, for the claim *"Amazon is doing great damage to tax paying retailers. Towns, cities, and states throughout the U.S. are being hurt - many jobs being lost!"*, it should be clear what is causing job loss in the second part. A possible split would be *"Amazon is doing great damage to tax paying retailers"* and *"Towns, cities and states throughout the U.S. are being hurt by Amazon - many jobs being lost"*. That is, it is necessary to rewrite the second part by adding *Amazon* a second time in order for the second part to be understandable without context.

### J.3.3 Claim Contextualization

Some claims are not complete, which means they lack adequate contextualization to be verified. For example, in the claim *"We have 21 million unemployed young men and women."*, there are unresolved pronouns without which the claim cannot be verified (e.g. *we* refers to Nigeria, as the speaker of the claim is the presidential candidate of Nigeria). Another example is *"Israel already had 50% of its population vaccinated."* We need to know when this claim was made to verify its veracity, as time is crucial for this verification. For the latter, metadata is enough to resolve ambiguities; the former needs to be rewritten as *"Nigeria has 21 million unemployed young men and women."*

Annotators are asked to contextualize claims to the original post by gathering the necessary information. Some information can be included simply as metadata, but this is not always enough – for information not captured by metadata, we ask that the claim itself is rewritten to include said information. Annotators need to follow this checklist:

1. Is the claim referring to entities that can only be identified by reading the associated fact-checking article, even if all metadata is taken into consideration? If so, add the names of the

entities (e.g. *"Former first lady said, 'White folks are what's wrong with America'."* becomes *"Former first lady Michelle Obama said, 'White folks are what's wrong with America'."*).

2. Does the claim have unnecessary quotation marks or references to a speaker (such as the word *says* in the example here)? If so, remove them (e.g. *"Says 'Monica Lewinsky Found Dead' in a burglary."* becomes *"Monica Lewinsky found dead in a burglary."*). Do NOT remove the reference to the speaker if the central problem is to determine if that person actually said the quote, e.g. in the case of quote verification.

3. Is the claim a question? If so, rephrase it as a statement (e.g. *"Did a Teamsters strike hinder aid efforts in Puerto Rico after Hurricane Maria?"* becomes *"A Teamsters strike hindered aid efforts in Puerto Rico after Hurricane Maria in 2017."*).

4. Does the claim contain pronominal references to entities only mentioned in the fact-checking article? If so, replace the pronoun with the name of that entity. (e.g. *"We have 21 million unemployed young men and women."* becomes *"Nigeria has 21 million unemployed young men and women."*).

5. For some fact-checking articles, the title used does not properly match the fact-checked claim. Find the original claim in the article, and use that for producing the normalized version. As shown in Figure 11, the claim should be the first sentence of the article rather than the title.

6. Is the claim too vague to be investigated through the use of evidence, and does the fact-checking article investigate a more specific version of the claim? If so, use the claim investigated in the fact-checking article (e.g. *"Towns, cities, and states throughout the U.S. are being hurt by Amazon"* might become *"Towns, cities, and states throughout the U.S. are losing state tax revenue because of Amazon"*).

Generally, try to make claims specific enough so that they can *be understood* and so that *appropriate evidence can be found* by a person who has not seen the fact-checking article.

**Important!** We recommend reading through the entire article and understanding the central problem before rewriting the claim. This makes it easier to identify the exact phrasing of the original claim and to make any minimal interventions necessary following our checklist above. When in doubt as to whether a claim should be modified, we recommend leaving it unchanged – we generally prefer claims to be as close as possible to their original form, subject to the constraints listed above.

### J.3.4  Labels

We ask annotators to produce a label for the claim relying *only* on the information on the fact-checking site (and assuming that everything reported it is accurate). For the dataset we are creating, we will be using four labels:

1. The claim is **supported**. The claim is supported by the arguments and evidence presented.

2. The claim is **refuted**. The claim is contradicted by the arguments and evidence presented.

3. There is **not enough evidence** to support or refute the claim. The evidence either directly argues that appropriate evidence cannot be found, or leaves some aspect of the claim neither supported nor refuted. We note that many fact-checking agencies mark claims as *refuted* (or similar), if supporting evidence does not exist, without giving any refuting evidence. We ask annotators to use *not enough evidence* for this category, regardless of the original label. In situations where evidence can be found that the claim is *unlikely*, even if the evidence is not conclusive, annotators may use *refuted*; here, annotators should use their own judgment. We give a few examples in Section J.3.5.

4. The claim is misleading due to **conflicting evidence/cherry-picking**, but not explicitly refuted. This includes cherry-picking (see `https://en.wikipedia.org/wiki/Cherry-picking`), true-but-misleading claims (e.g. the claim *"Alice has never lost an election"* with evidence showing Alice has only ever run unopposed), as well as cases where conflicting or internally contradictory evidence can be found.

   Conflicting evidence may also be relevant if a situation has recently changed, and the claim fails to mention this (e.g. *"Alice is a strong supporter of industrial subsidies"* with evidence showing that Alice currently supports industrial subsidies, but in the past opposed industrial

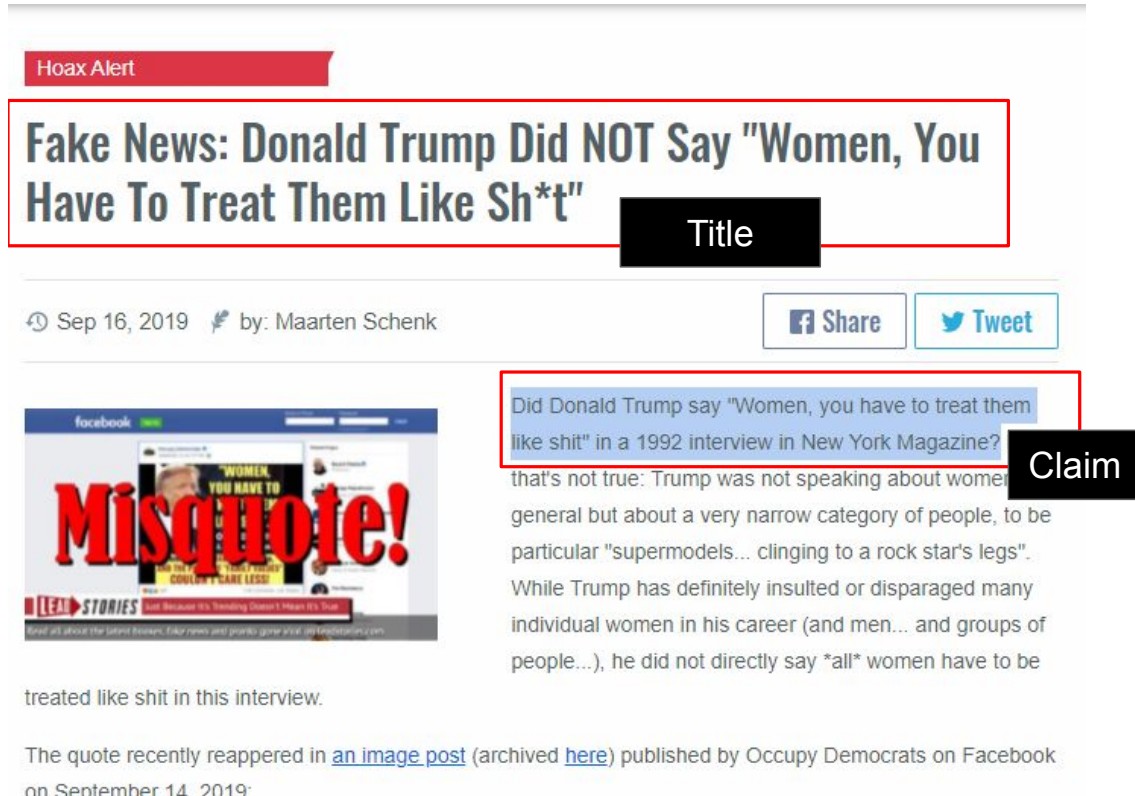

Figure 11: An example of locating the claim.

subsidies). We note that if the claim covers a period of time, and evidence refutes the claim at some timepoints but not others, the whole claim is still *refuted* – for example, *"Alice has always been a strong supporter of industrial subsidies"* or *"Alice has never been a strong supporter of industrial subsidies"*. For a real example from our dataset, consider `https://fullfact.org/online/does-polands-migration-policy-explain-its-lack-terror-attacks/` – the claim is that *"Poland has had no terror attacks"*; evidence shows that Poland had no terror attacks before 2015, but some examples afterward, and should as such be marked *refuted*.

Despite the claim splitting subtask, some claims may contain multiple parts that are too interconnected to split. This could for example be a claim like *"Alice has never lost an election because she always supports cheese subsidies"*. In such cases, parts of the claim may have different truth values. We discuss a few cases below:

- The claim is implicature, i.e. *"X happens because Y"* or *"X leads to Y"*. In this case, annotators should find a label for the causal implication, and *not* for either of the component claims.

- The claim has too components, where one is *refuted* and the other is *not enough information*. In this case, the entire claim should be labeled *refuted*.

- The claim has too components, where one is *supported* and the other is *not enough information*. In this case, the entire claim should be labeled *not enough information*.

**Important!** The label was given in Phase 1 – and *only* in Phase 1 – should reflect the decision of the fact checker, not the interpretation of the annotator. In Phase 1, annotators should report the original judgment, as closely as possible, even if they disagree with it.

### J.3.5 Deciding Between Refuted and NEE

As mentioned, the line between *refuted* and *not enough evidence* requires annotators to rely on their own judgment in cases where refuting evidence cannot be directly found, but the claim is extremely unlikely. As a guiding principle, if annotators would feel doubt regarding the truth value of the claim – given the presented evidence and/or lack of evidence – *not enough evidence* should be chosen. Below, we give several examples from our dataset:

- *"The Covid-19 dusk-to-dawn curfew is Kenya's first-ever nationwide curfew since independence."* No evidence can be found that Kenya has implemented a nationwide curfew before the Covid-19 pandemic. However, it is conceivable that evidence of such a curfew would simply not show up in documentation uploaded to the internet. As such, the annotator cannot rule out a prior curfew beyond a reasonable doubt, and such should select *not enough evidence* as the label.

- *"The government in India has announced that it will shut down the internet to avoid panic about the Coronavirus."* Evidence can be found that Indian law allows the government to do so as an emergency measure; however, the annotator finds no announcement from the government that the internet actually will be shut down. If other, regular, announcements from the same government body could be found, the claim should be labeled *refuted* – it would be extremely unlikely that a shutdown on the internet would not be announced via standard channels. However, in this case, standard channels do not make announcements in English, and therefore it is plausible that the announcement has not been found simply because it has not been translated; in this case, the annotator should select *not enough evidence* (with evidence that no English-language official channel exists).

- *"Shakira is Canadian."* Evidence can be found that Shakira is usually described as Colombian, was born in Colombia, and holds Colombian citizenship. Furthermore, evidence shows she now resides in Spain. As no evidence of any connection to Canada can be found despite the wealth of information available about her, it is extremely unlikely that she is secretly Canadian; as such, the annotator can select *refuted* as the label.

A special case of this kind of claim is quote verification, where it can be difficult to establish that someone did *not* say something. In many cases, evidence can be found that a quote is fictional (e.g. by finding evidence from a service like `https://quoteinvestigator.com/`), or that it originates from someone else. However, in some cases, there is no readily available evidence. In this case, we advise that annotators document the lack of evidence that the person said *the quote itself*, or *any paraphrase of the quote*. Further, annotators should document that *some* quotes by that person can be found, if possible what the person has said *on the same topic*, and if possible that the quote has not been said by *someone else*. This establishes that evidence for the quote should be available, and is not; in that case, annotators can pick *refuted* as the label. If annotators cannot find any claims by the person or any evidence for the quote (say an entirely fictional person with an entirely fictional quote), they should pick *not enough evidence*.

For a good example of how to handle these cases, consider the claim *"RBI has said that ₹2000 notes are banned and ₹1000 notes have been introduced"*. As this claim is false, no evidence can be found of RBI making any such announcement; nor that they did *not* make that particular announcement. Here, the annotator first established where official communication from RBI is published with the question *"how do the RBI/central bank make announcements on changes to currency?"* Then, after finding that all official communication is posted to the RBI website, they asked a follow-up question testing whether evidence for the claim can be found *on the official website*.

### J.3.6 Metadata Collection

Annotators need to collect metadata through the following three steps.

### J.3.7 General Information

- A hyperlink to the original claim, if that is provided by the fact-checking site. Examples of this include Facebook posts, the original article or blog post being fact-checked, and embedded video links. If the original claim has a hyperlink on the fact-checking site, but that hyperlink is dead, annotators should leave the field empty.

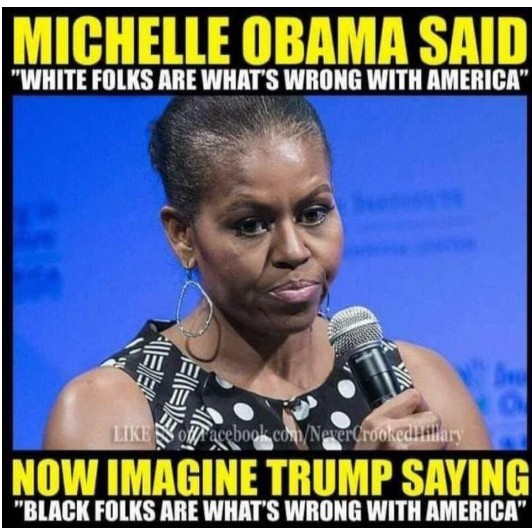

Figure 12: An example of an image claim requiring transcription.

- The date of the original claim, regardless of whether it is necessary for verifying the claim. This date is often mentioned by the fact checker, but not in a standardized place where we could automatically retrieve it. Note that the date for the *original claim* and the *fact-checking article* (often its publication date) may be different and both are stated in the text. We specifically need the original claim date, as we intend to filter out evidence that appeared after that date. If multiple dates are mentioned, the earliest should be used. If an imprecise date is given (e.g. February 2017), the earliest possible interpretation should be used (i.e. February 1st, 2017).

- The speaker of the original claim, e.g. the person or organization who made the claim.

- The source of the original claim, e.g. the person or organization who published the claim. This is not necessarily the same as the speaker; a person might make a comment in a newspaper, in which case the person is the speaker and the newspaper is the source.

- If the original claim is or refers to an image, video, or audio file, annotators should add a link to that media file (or the page that contains the file, if the media file itself is inaccessible).

- If the original claim is an image that contains text – for example, Figure 12 shows a Facebook meme about Michelle Obama – annotators should transcribe the text that occurs in the image as metadata. In the example, it would be *"Michelle Obama said white folks are what's wrong with America."*

- If the fact-checking article is paywalled or inaccessible due to an error message, annotators should report this and skip the claim using the corresponding button.

### J.3.8 Claim Type

The type of the claim itself, independent of the approach taken by the fact checker to verify or refute it, should be chosen from the following list. This is not a mutually exclusive choice – a claim can be speculation about a numerical fact, for example. As such, annotators should choose one *or several* from the list.

- **Speculative Claim**: The primary task is to assess whether a prediction is plausible or realistic. For example *"the price of crude oil will rise next year."*

- **Opinion Claim**: The claim is a non-factual opinion, e.g. *"cannabis should be legalized"*. This contrasts with factual claims on the same topic, such as *"legalization of cannabis has helped reduce opioid deaths."*

- **Causal Claim**: The primary task is to assess whether one thing caused another. For example *"the price of crude oil rose because of the Suez blockage."*.

- **Numerical claim**. The primary task is to verify whether a numerical fact is true, or to verify whether a comparison between several numerical facts hold, or to determine whether a numerical trend or correlation is supported by evidence.

- **Quote Verification**. The primary task is to identify whether a quote was actually said by the supposed speaker. Claims *only* fall under this category if the quote to be verified directly figures in the claim, e.g. *"Boris Johnson told journalists 'my favourite colour is red, because I love tomatoes' "*.

- **Position Statement**. The primary task is to identify whether a public figure has taken a certain position, e.g. supporting a particular policy or idea. For example, *"Edward Heath opposed privatisation"*. This also includes statements that opinions have changed, e.g. *"Edward Heath opposed privatisation before the election, but changed his mind after coming into office"*. Factual claims about the actions of people (e.g. *"Edward Heath nationalised Rolls-Royce"*) are not position statements (they are event or property claims); claims about the attitudes of people (e.g. *"Edward Heath supported the nationalisation of Rolls-Royce"*) are.

- **Event/Property Claim**. The primary task is to determine the veracity of a narrative about a particular event or series of events, or to identify whether a certain non-numerical property is true, e.g. a person attending a particular university. Some properties represent causal relationships, e.g. *"The prime minister never flies, because he has a fear of airplanes"*. In those cases, the claim should be interpreted as both a property claim and a causal claim.

- **Media Publishing Claim**. The primary task is to identify the original source for a (potentially doctored) image, video, or audio file. This covers both doctored media, and media that has been taken out of context (e.g. a politician is claimed to have shared a certain photo, and the task is to determine if they actually did). This also includes HTML-doctoring of social media posts. We will discard all claims in this category.

- **Media Analysis Claim**. The primary task is to perform complex reasoning about pieces of media, distinct from doctoring. This could for example be checking whether a geographical location is really where a video was taken, or determining whether a specific person is actually the speaker in an audio clip. The claim itself *must directly involve* media analysis; e.g. "the speaker of these two clips is the same". Claims where the original source is video, but which can be understood and verified without viewing the original source, do not fall under this category. An original video or audio file can feature as metadata in fact-checking articles, but claims are only *complex media claims* if analysis of the video or audio beyond just extracting a quote is necessary for verification.

Several claim types – speculative claims, opinion claims, media publishing claims, and media analysis claims – will not be included in later phases.

### J.3.9 Fact-checking Strategy

After identifying the claim type, we ask annotators to classify the approach taken by the fact checker according to the article. This is independent of the claim type, as a fact-checker might take any number of approaches to a given claim. Again, one *or several* options should be chosen from the following list:

- **Written Evidence**. The fact-checking process involved finding contradicting or supporting written evidence, e.g. a news article directly refuting or supporting the claim.

- **Numerical Comparison**. The fact-checking process involved numerical comparisons, such as verifying that one number is greater than another.

- **Consultation**. The fact checkers directly reached out to relevant experts or people involved with the story, reporting new information from such sources as part of the fact-checking article.

- **Satirical Source Identification**. The fact-checking process involved identifying the source of the claim as satire, e.g. The Onion.

- **Media Source Discovery**. The fact-checking process involved finding the original source of a (potentially doctored) image, video, or soundbite.

- **Image analysis**. The fact-checking process involved image analysis, such as comparing two images.

- **Video Analysis**. The fact-checking process involved analysing video, such as identifying the people in a video clip.

- **Audio Analysis** The fact-checking process involved analysing audio, such as determining which song was played in the background of an audio recording.

- **Geolocation**. The fact-checking process involved determining the geographical location of an image or a video clip, through the comparison of landmarks to pictures from Google Streetview or similar.

- **Fact-checker Reference**. The fact-checking process involved a reference to a previous fact-check of the same claim, either by the same or a different organisation. Reasoning or evidence from the referenced article was necessary to verify the claim.

Claims *only* labelled as solved through Fact-checker Reference will not be included in later phases.

### J.4 Phase 2: Question Generation and Answering

The next round of annotation aims to produce pairs of questions and answers providing evidence to verify the claim. The primary sources of evidence are the URLs linked in the fact-checking article. We also provide access to a custom search bar to retrieve evidence.

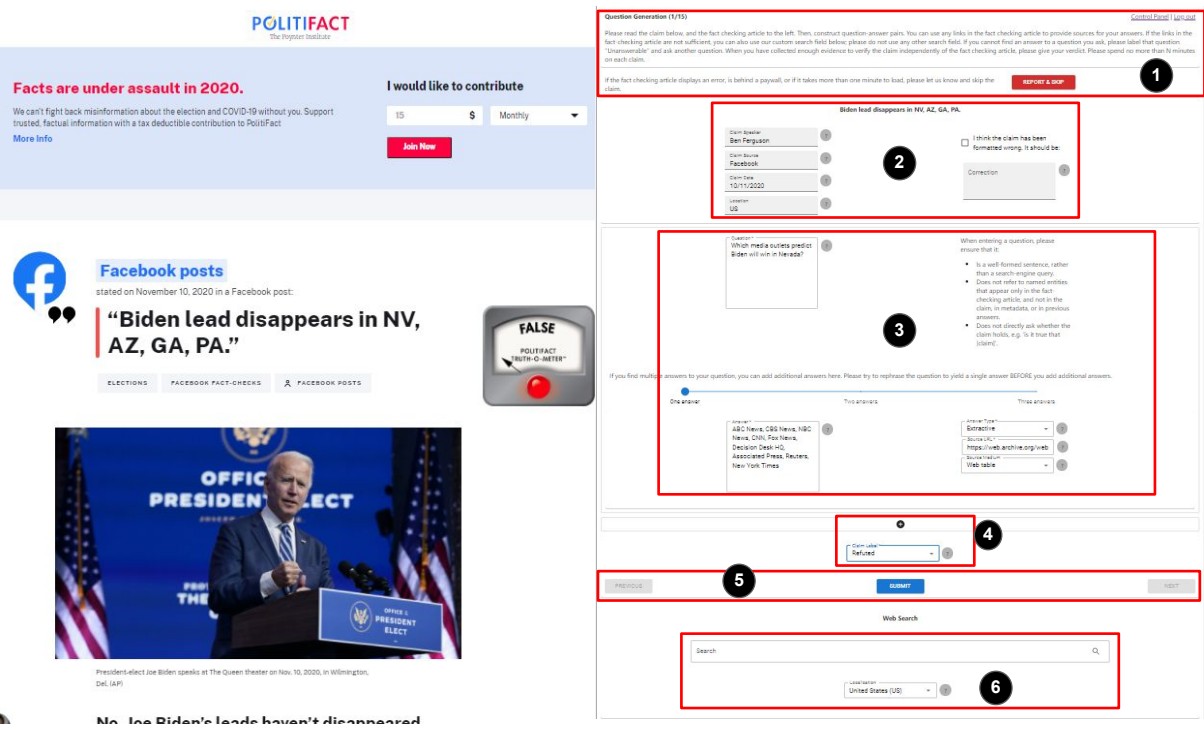

Figure 13: Interface of question generation. (1) Guideline of annotation for this phase. Please read it before annotating. Notice that if the article displays a 404 page or another error, or if it takes more than one minute to load, please click the **REPORT & SKIP** button. (2) The claim and the associated metadata. (3) Fields for the first question and its answers. Annotators can add up to 3 answers for each question if necessary. The text fields of metadata of question answer pairs are also provided. (4) Annotators can use the plus button to add as many questions as they want. Please select the label of this claim after finishing the question and answer generation. (5) Buttons for submitting the current claim, going to the previous claim, and next claim. (6) The custom search engine.

### J.4.1 Overview

Here, we give a quick overview of the question generation task; in-detail discussion can be found in subsequent sections. Further documentation can also be found on-the-fly using the tooltips in the annotation interface.

1. The annotator should first read the claim and metadata provided by the previous annotator, and the associated fact-checking article (including the verdict). We note that because phase-one annotators sometimes split decompose claims into parts, in some cases not all sections of the fact-checking article will be relevant.

2. The task is then to generation questions and answers about the claim such that a verdict can be given without knowledge of the fact-checking article. The sources and strategies used in the fact-checking article can serve as inspiration for questions and evidence for answers, but the fact-checking article should not be *directly* referenced as a source.

3. If an annotator believes a phase one claim has been extracted wrongly, they can correct it using the appropriate box. This is not necessary for most claims, but adds an extra layer of quality control. Guidance on correcting claims along with examples can be found in Section J.4.2.

4. We recommend constructing question-answer pairs iteratively, one at a time. That is, annotators should ask a question and attempt to answer it, and only then proceed to the next question.

5. Guidance on generating questions can be found in Section J.4.3.

6. Answers should be sought from the metadata, any of the sources listed on the fact-checking article (e.g. any hyperlinks to other sites), and when that is not possible (e.g. due to the hyperlinks being dead) from the internet using the search bar we provide.

7. Questions about metadata can be used to draw attention to aspects of the claim, in order to reason about publication date or publication source (see Section J.4.4).

8. WARNING: For persistence, we have stored all fact-checking articles on archive.org. Fact-checking articles may feature double-archived̈links using both archive.org and archive.is, e.g.
`https://web.archive.org/web/20201229212702/https://archive.md/28fMd`.
Archive.org returns a 404 page for these. To view such a link, please just copy-paste the archive.is part (e.g.
`https://archive.md/28fMd`) into your browser.

9. Answers should be accompanied by a hyperlink to the source, and the type of the source – e.g. web text, a pdf – should be specified. We note that if the source type is set as metadata, the source link will automatically be set to the word *metadata*.

10. Answers can be either *extraction*, e.g. copy-pasted directly from the source, *abstractive*, e.g. written in free-form based on the source, or *boolean*, e.g. written as yes/no with an explanation taken either extractively or abstractively from the source. Where possible, we strongly prefer extractive answers.

11. If an answer cannot be found, we also allow annotators to mark the question as unanswerable. We ask annotators to use this instead of deleting unanswerable questions.

12. Guidance on generating answers can be found in Section J.4.6.

13. If enough questions have been asked to support a verdict, or if at least ten minutes have passed without the annotator finding enough evidence, a verdict should be given from our for labels described in Section J.3.4.

14. Annotators in phase two should base their verdict on the question-answer pairs they have generated, and *not* on the fact-checking article. Depending on what information has been retrieved, they may therefore disagree with the article.

15. Before proceeding to the next hit, the annotator will be shown a warning with the QA-pairs they have generated. They will also be shown their assigned label. They will be asked to confirm that the collected evidence is sufficient to assign the label they have chosen to the claim.

16. Sometimes, annotators may be in doubt as to whether an additional question should be added to further support the verdict. Generally speaking, we always prefer to have as many question-answer pairs as possible, so if in doubt annotators should veer on the side of adding that additional question.

**Important!** Annotators should not choose a label if the retrieved evidence does not support it; for example, if the label **conflicting evidence** is chosen, there should be evidence documenting the conflict. Labels in phase two can contradict the label of the fact-checker, if the annotator believes it is appropriate.

### J.4.2 Claim Correction

In addition to gathering question-answer pairs, Phase Two also acts as quality control for the claim contextualization in Phase One. This means if Phase Two annotators encounter a claim that is malformed or not properly contextualized, they can correct it. The guidelines for claim contextualization can be seen in Section J.3.3; the same criteria hold. Based on our initial review of the data entered in Phase One, Claim Correction is rarely necessary. Below are some examples from the data of claims that *should* be corrected in Phase Two:

1. The claim *"Nigerian vice presidential candidate Peter Obi claimed that Capital expenditure in 2016 was N1.2 trillion and 2017 was N1.5 trillion."*, given the article `https://afri cacheck.org/fact-checks/reports/battle-titans-fact-checking-arch-riv als-race-nigerias-presidency`. The article verifies the numerical value of capital expenditure in Nigeria, not whether Peter Obi has claimed anything about it. The original article is not quote verification, but the annotator has changed the claim to that. Here, the Phase Two annotator should correct the claim to simply *"Nigerian capital expenditure in 2016 was N1.2 trillion and 2017 was N1.5 trillion."*

2. The claim *"Abolish all charter schools"*, given the article `https://www.factcheck.or g/2020/07/trump-twists-bidens-position-on-school-choice-charter-sch ools/`. This is a position statement about Joe Biden's stance on charter schools; however, the annotator has removed all reference to Joe Biden. The Phase Two annotator should correct the claim to *"Joe Biden wants to abolish all charter schools"*.

3. The claim *"Is Florida doing five times better than New Jersey?"*, given the article `https://leadstories.com/hoax-alert/2020/07/fact-check-florida-is-not-doing -five-times-better-in-deaths-than-new-york-and-new-jersey.html`. The claim has mistakenly been phrased as a question. It is also too vague. The Phase Two annotator should correct this, following the article: *"Florida is doing five times better than New Jersey in COVID-19 deaths per 1 million population"*.

### J.4.3 Question Generation

To ensure the quality of the generated questions, we ask the annotators to create their questions as follows:

- Questions should be well-formed, rather than search engine queries (e.g. "where is Cambridge?" rather than "Cambridge location").
- Questions should be standalone and understandable without any previous questions.
- Questions should be based on the version of the claim shown in the interface (i.e. the version extracted by phase one annotators), and not on the version in the fact-checking article. If an annotator believes a phase one claim has been extracted wrongly, they can correct it using the appropriate box.
- The annotators should avoid any question that directly asks whether or not the claim holds, e.g. *"is it true that [claim]"*.
- The annotators should ask all questions necessary to gather the evidence needed for the verdict, including world knowledge that might seem obvious, but could depend for example on where one is from. For example, Europeans might have better knowledge of European geography/history than Americans, and vice-versa.

- As a guiding principle, at least 2 questions should be asked. This is not a hard limit, however, and the annotators can proceed with only one question asked if they do not feel more are needed.

The following are examples used to illustrate how questions should be asked. These are based on the real claim *"the US in 2017 has the largest percentage of immigrants, almost tied now with the historical high as a percentage of immigrants living in this country"*:

- Good: What was the population of the US in 2017?
- Good: How many immigrants live in the US in 2017?
- Bad: What was the population of the US? [No time specified to find a statistic]
- Bad: What was the population there in 2017? [What does *there* refer to?]
- Bad: Is it true that the US in 2017 has the largest percentage of immigrants, almost tied now with the historical high as a percentage of immigrants living in this country? [Directly paraphrases the claim]

### J.4.4 Metadata

Questions about metadata can be used to draw attention to aspects of the claim, in order to reason about publication date or publication source. If, for example, the claim *"aliens made contact with earth March 3rd, 2021"* was published on September 1st, 2020, the publication date can be used to refute the claim. In such cases, we ask annotators to first generate a question/answer pair – *"when was this claim made?" "September 1st, 2020"* – which can then be used to refute the claim. Similarly, questions about publication source can be used to refute satirical claims – *"where was this claim published?"*, *"www.theonion.com"*, *"what is The Onion?"*, *"The Onion is an American digital media company and newspaper organization that publishes satirical articles on international, national, and local news."*.

### J.4.5 Common sense assumptions and world knowledge

As a part of the question generation process, annotators may have to make assumptions and/or use world knowledge to interpret the claim. For example, for the claim *"Shakira is Canadian"*, it may be necessary to choose what it means to be Canadian. This is expressed in how questions are formulated, e.g. *"does Shakira have Canadian citizenship?"* or *"where does Shakira live?"*. This may also involve politically charged judgments. For example, some First Nations people are classed as Canadian by the Canadian government, but do not use that label for themselves.

In such cases, we ask annotators to follow – as closely as possible – the judgments made by the fact-checking websites. If the annotators feel that these are incomplete or misleading, they can add additional questions.

For example, for the claim *"Edward Heath opposed privatisation"*, a fact checker might provide his party manifesto as evidence. A corresponding question could then be *"what did the 1970 Conservative Party manifesto say about privatisation?"* An annotator could encounter evidence for the nationalisation of Rolls Royce during Heath's government, which the fact-checking article did not take into account. In that case, the annotator might want to add an additional question, such as *"did Heath's government nationalise any companies?"*. The annotators should ask *both* questions.

**Important!** As opposed to Phase 1, annotators in Phase 2 *should* use their own judgment to assign labels (although they should not ignore evidence used by the fact-checker). As such, if they disagree with the fact-checker about the label, they can select a different label.

### J.4.6 Answer Generation

To find answers to questions, the annotators can rely on metadata, or on any sources linked from the factchecking site. Where these fail to produce appropriate information – either because they are not relevant to an asked question or because they refer to sources which have been taken down – we provide search functionalities as an alternative. Note that the annotators are not allowed to use the fact-checking article itself as a source, only the pages *hyper-linked* in the fact-checking article (and

only when they are not from fact-checking websites). Similarly, other fact-checking articles found through search should be avoided.

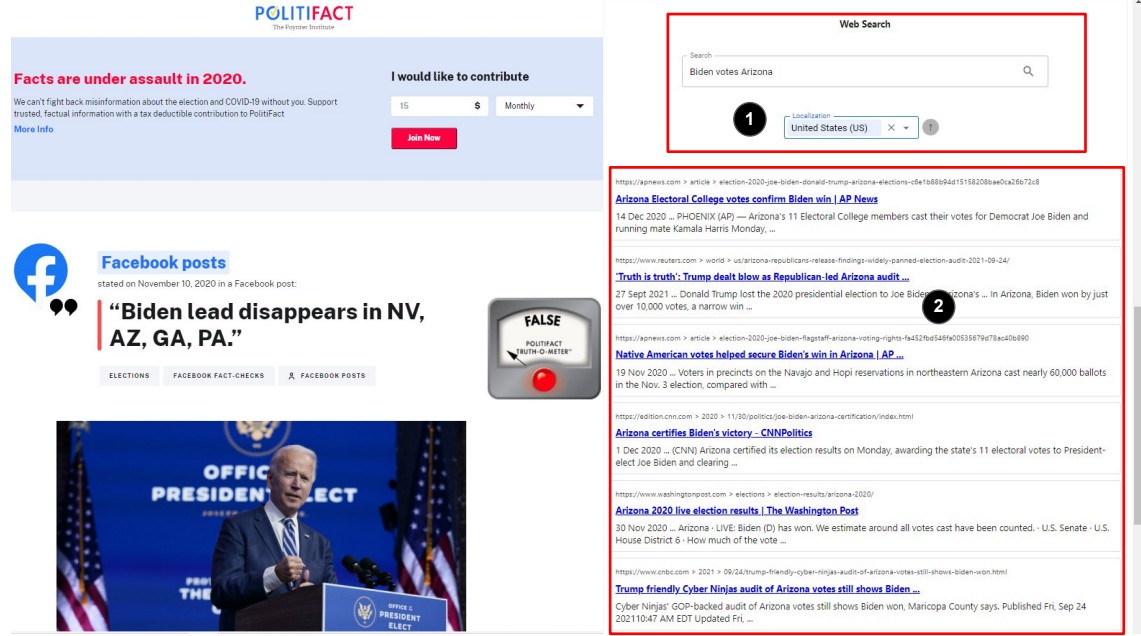

Figure 14: Interface of the search bar. (1) Search bar and the location option. Annotators can change the localization of the search engine by selecting the country code here. (2) Search results returned by the search engine.

Once an answer has been found, annotators can choose between the following four options to enter it:

- **Extractive:** The answer can be copied directly from the source. We ask the annotators to use their browser's copy-paste mechanism to enter it.
- **Abstractive:** A freeform answer can be constructed based on the source, but not directly copy-pasted.
- **Boolean:** This is a special case of abstractive answers, where a yes/no is sufficient to answer the question. A second box must be used to give an explanation for the verdict grounded in the source (e.g. "yes, because...").
- **Unanswerable:** No source can be found to answer the question.

For extractive, abstractive, and boolean answers, the annotators are also asked to copy-paste a link to the source URL they used to answer the question. Extractive answers are preferred to abstractive and boolean answers.

In some cases, annotators might find different answers from different sources. Our annotation tools allows adding additional answers, up to three. While we provide this functionality, we ask that annotators try to rephrase the question to yield a single answer before adding additional answers.

We note that if the annotators can only find a *partial* answer to a question, they can still use that. In such cases, please give the partial answer rather than marking the question as unanswerable.

Our search engine marks pages originating from known sources of misinformation and/or satire. We do not prevent annotators from using such sources, but we ask that annotators avoid them if at all possible. In the event that an annotator wishes to use information from such a source, we strongly prefer that the finds similar, corroborating information from an additional source in order to further substantiate the evidence.

While answering a question, we furthermore ask annotators to adhere to the following:

**Important!**

- DO NOT use any other browser window/search bar to find an answer. You MUST use the provided search bar only.

- DO NOT give a verdict for the claim until you have finished questions and answers.

- DO NOT use the fact-checking article itself, or any other version of it you find on the internet, as evidence to support an answer.

- DO NOT submit answers using other articles from fact-checking websites, such as politifact.com or factcheck.org, as evidence.

- DO NOT simply reference the source as an authority in abstractive answers (and boolean explanations), e.g. do not use answers like *"yes, because the Guardian says so"*. Rather, write out what the source says, e.g. *"yes, because £18.1 bn is 41% of the budget"*. If you consider it important to mention the source, write that the source says – e.g., *"yes, because according to the Guardian £18.1 bn was spent, which is 41% of the budget"*.

### J.4.7 Reasoning Chains of Claims

Annotators can build up reasoning chains across multiple questions, meaning that answers of one question can be used in the next question. For example, for the claim *"the fastest train in Japan drives at a top speed of 400 km/h"*, the first question is "What is the fastest Japanese train?". The answer is "The fastest Japanese train is Shinkansen ALFA-X". Based on the answer, we can further ask the second question to get more details, "What is the maximum operating speed of the Shinkansen ALFA-X". Note that while the *generation* of the second question assumes knowledge of the answer to the first, it is *understandable* without it.

### J.4.8 Confirmation

After submitting the question/answer pairs for a claim, annotators will be presented with a confirmation screen (see Figure 15). Annotators will be shown the question/answer pairs they have entered, along with the verdict, and asked to confirm a second time that the verdict is supported by the evidence.

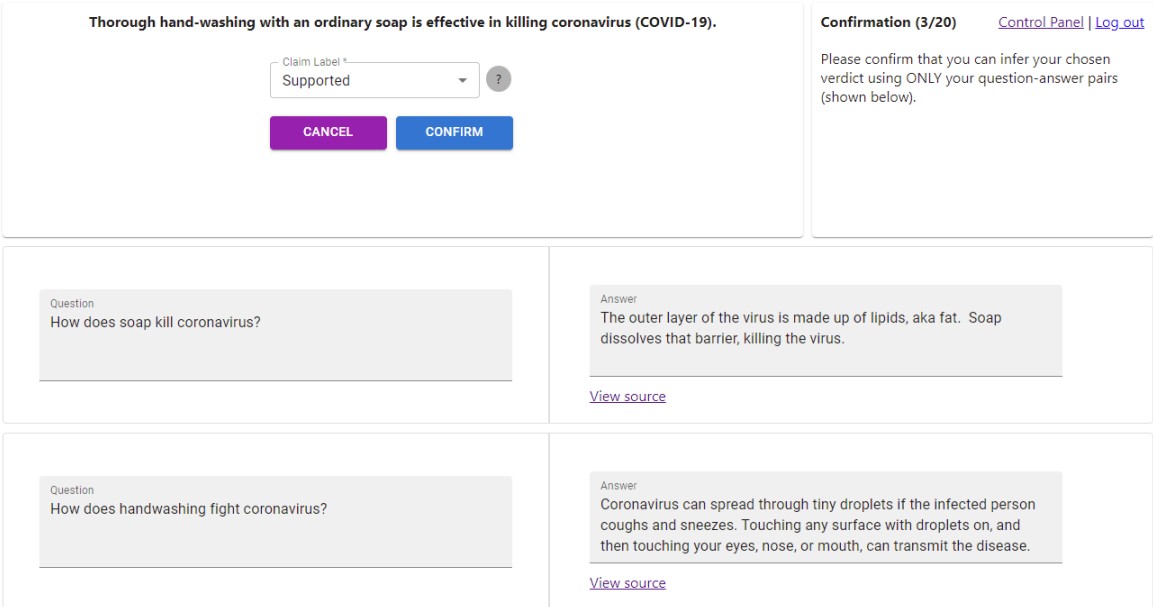

Figure 15: Before moving on to the next claim, phase two annotators will be shown a confirmation screen to make sure that their chosen verdict is correct.

### J.5 Phase 3: Quality Control

Once we have collected evidence in the form of generated questions and retrieved answers, we want to provide a measure of quality. Given a claim with associated evidence, we ask a third round of annotators to give a verdict for the claim. Crucially, the annotators at this round do not have access to the original fact-checking article, or to the claim label.

#### J.5.1 Overview

Here, we give a quick overview of the quality control task; in-detail discussion can be found in the following sections. Further documentation can also be found on-the-fly using the tooltips in the annotation interface.

1. Annotators should first read the claim, the metadata, and the question-answer pairs. This is the only information which should be used during this phase

2. It is important that annotators in the quality control phase do not use web search to find additional information, or rely on background knowledge which an average English speaker might not have. Commonsense facts that are known to (almost) everyone can be used – see Section J.5.2.

3. If the claim, or any of the question-answer pairs lack context, they can be flagged. This helps us diagnose what is wrong with a set of question-answer pairs in the case annotators disagree over the label.

4. After reviewing the claim and the QA pairs, annotators should assign a label to the claim (see the four labels introduced in Section J.3.4).

5. Finally, annotators should write a short statement justifying the verdict. If any commonsense information (e.g. background knowledge which an average English speaker *is* likely to have) is used to give the verdict, but that information is not mentioned in any question-answer pair, it should be mentioned in the justification. For advice regarding justification production, see Section J.5.3.

#### J.5.2 Commonsense Knowledge

When giving a verdict, annotators sometimes need to rely on commonsense knowledge. Here, we consider only basic facts which an average English speaker is likely to know – e.g. *"Earth is a planet"* or *"raindrops consist of water"*. No other information beyond the question-answer pairs can be used in this phase.

We ask annotators to be relatively strict with what they consider commonsense, but use their own judgment. For example, we would consider *"Canada is a country"* commonsense, but not *"Canada is the third-largest country in terms of land mass"*. If an annotator is in doubt as to whether something is considered commonsense, they should not consider it commonsense.

#### J.5.3 Justification Production

In addition to the verdict, we as mentioned also ask annotators in Phase Three to write a short statement justifying their verdict. This justification should explain the reasoning process used to reach the verdict, along with any commonsense knowledge. If calculations or comparisons were used, e.g. *"6.3% is greater than 6.1%"* or *"10-4=6"*, they should be explicitly stated in the justification. Similarly, any rounding logic – e.g. *"4.3 million is approximately 4 million"* – should be explicitly stated here.

Other than commonsense knowledge, there should not be any new information presented in this statement. The justification should only describe how the annotators used the information present in the claim, the metadata, and the QA-pairs to reach their verdict. If a verdict cannot be reached, e.g. if the *not enough information*-label is chosen, annotators should instead describe what information is missing – e.g. *"I cannot determine if Canada is the third-largest country, because the questions do not specify how large any countries are."*

Similarly, in cases of conflicting evidence, annotators should describe which questions and answers lead to the conflict, and how they contradict – e.g. *"This claim is cherry-picked as it looks only at the*

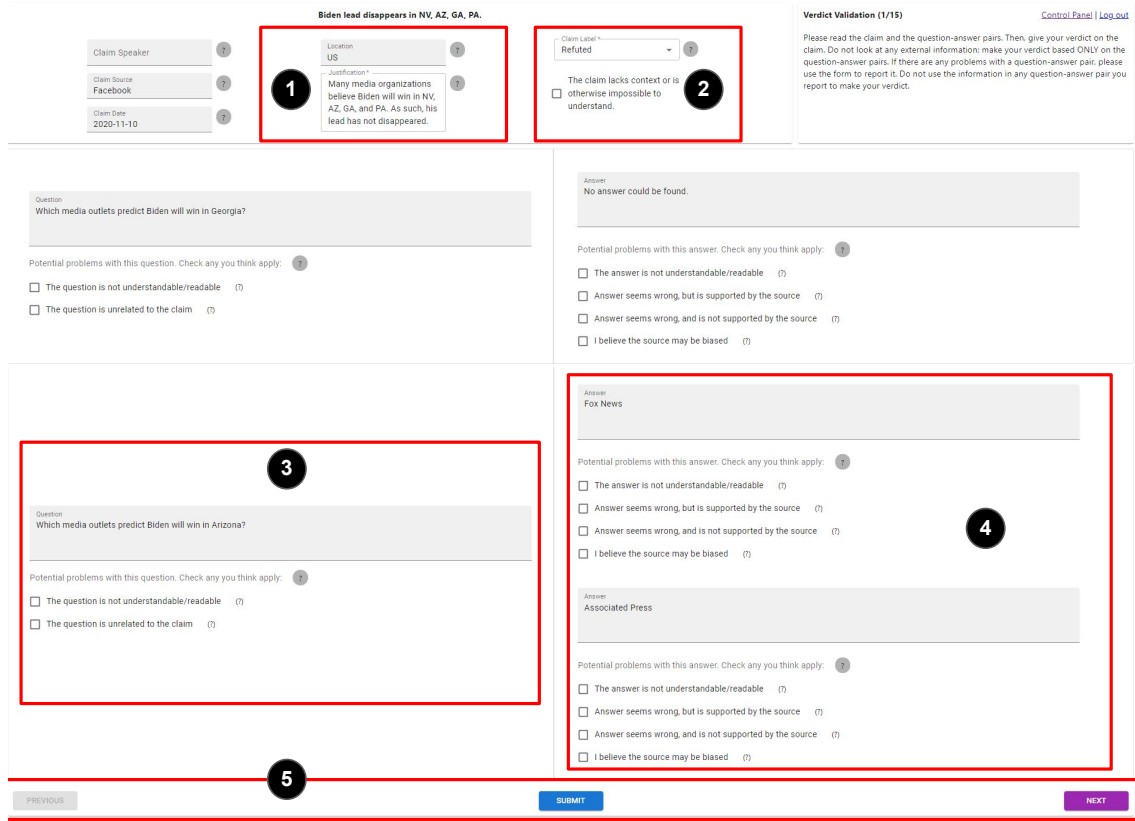

Figure 16: Interface of quality control. ①Text field for entering the justification. ②Label of the claim and the checkbox of unreadable. Notice that once the unreadable option is selected, annotators do not need to select the label for the claim. ③The question corresponds to the current claim. Here we have two question-answer pairs. If the annotator think the there exist potential problems with this question, check any options applied. ④The answers corresponds to the question on the left. If the annotator think the there exist potential problems with the answer, check any options applied. ⑤Buttons for submitting the current claim, going to the previous claim, and next claim.

*price of vanilla icecream, for which an increase did take place, but leaves out other flavours, where no increase happened."*

## J.6 Dispute Resolution

For some claims, there may be a disagreement between the labels produced by annotators in the question generation and quality control phases. In those cases, the claim will go through a second round of question generation and quality control. While the instructions given in Sections J.4.3 and J.5 still apply, we give a few extra recommendations specific to dispute resolution here.

### J.6.1 Vague Claims

Some claims may pass to the dispute resolution phase because they are too vague for annotators in phases two and three to agree on the meaning. In order to catch these cases, the final step of dispute resolution – that is, the extra quality control step at the end – includes an additional label, *Claim Too Vague*. This should be select when and only when an annotator can understand the claim (e.g. it is readable), but there is too much doubt over how it is supposed to be interpreted. For example, the claim *"Ohio is the best state"* is too vague as it is not clear what "best" refers to.

### J.6.2 Adding and Modifying Questions

The aim of dispute resolution is to resolve the conflict so that a potential new reader would come to a conclusive verdict. As such, the annotator should not necessarily agree with either the Phase Two or the Phase Three-verdict; they should attempt to make the fact-checking unambiguous. There may be cases where new questions must be added, and cases where existing questions should be changed but no new questions are necessary. There may also be cases where no change to the evidence is necessary at all, but where either the Phase Two or Phase Three-annotator has simply entered a wrong verdict. For this final category adding additional evidence to provide clarity can still be helpful, but it is not necessary; annotators should use their own judgment here.

### J.6.3 NEI-verdicts

A common case for dispute resolution is the situation where the Phase Two annotator has selected *Supported*, *Refuted*, or *Conflicting Evidence/Cherrypicking* as the verdict, but the Phase Three annotator has selected *Not Enough Evidence*. This can happen for example if Phase Two annotators forget to gather some of the evidence they use to reach the verdict, rely on aspects only stated in the fact-checking article without making it explicit through a question-answer pair, or overestimate the strength of the evidence they have gathered. In these cases, the aim of dispute resolution is to gather additional evidence and resolve the conflict that way; i.e. it is not sufficient to give a *Not Enough Information*-verdict without attempting to add evidence (although the same time limit as in P2 applies).

