# OpenReview forum: "AVeriTeC: A Dataset for Real-world Claim Verification with Evidence from the Web"
_NeurIPS.cc/2023/Track/Datasets_and_Benchmarks — NeurIPS 2023 Datasets and Benchmarks Poster_

### Official Review · Reviewer_rgH7 · 2023-07-21
**Comprehensive Dataset with Significant Improvement over Prior Benchmarks**

**Rating:** 8
**Confidence:** 3
**Correctness:** Yes.
**Clarity:** Yes.

**Strengths:**

- Comprehensive but refined dataset that improves upon similar existing datasets. Specifically, a much richer claim format with question-answer pairs.
- Robust annotation process filtering out low-quality evidence annotation
- Interesting and reasonable approach to automated evaluation of evidence-based fact-checking
- Sound empirical analysis

**Additional Feedback:**

Are there any plans to extend your evaluation method?

**Documentation:**

Yes.

**Ethics:**

No.

**Limitations:**

Yes.

**Opportunities For Improvement:**

- Evaluation method cannot handle alternative lines of justification

**Relation To Prior Work:**

Yes.

**Summary And Contributions:**

The authors introduce AVeriTeC, a diverse and comprehensive dataset of real-world claims annotated with question-answer pairs derived from online evidence. The dataset is sourced via a robust annotation process, filtering out low-quality claims with insufficient evidence. Alongside this dataset, the authors also introduce a novel, automated evaluation scheme for verifying claims. They benchmark multiple LLMs using this scheme on their dataset.

---

> ### Author Response · Authors · 2023-08-17
>
> We are thrilled to hear the reviewer finds our dataset comprehensive, interesting, and sound.
>
> **Evaluation with alternative lines of justification**
>
> While we agree that this is an issue – we acknowledge as much in the limitations section of our paper – it would actually be possible to use our evaluation metric with multiple lines of justification. Following the comments of Reviewer 3, we have improved our section on evaluation. We have added a section outlining how our technique could be applied in a multi-reference setting, similar to evaluation in e.g. Machine Translation.

---

### Official Review · Reviewer_GQze · 2023-07-21
**A very good paper for this track.**

**Rating:** 9
**Confidence:** 4
**Correctness:** Yes
**Clarity:** Yes

**Strengths:**

•	Very well structured and written paper, including the attachments.
•	The area of automatic claim verification is a hot topic.
•	The collection and annotation of a dataset is done very professionally, with care to all details.
•	Created a new real-world fact-checking dataset consisting of 4,568 claims, each annotated with question-answer pairs decomposing the fact-checking process, as well as justifications.
•	A multi-step annotation process is used, which guarantees high-quality annotations.
•	A baseline and an evaluation scheme, are proposed.
•	The new dataset is suitable as a new benchmark.


**Additional Feedback:**

-

**Documentation:**

yes.

**Ethics:**

Not

**Limitations:**

Yes.

**Opportunities For Improvement:**

If it is possible, increase the font size in Figure 2.

**Relation To Prior Work:**

fine

**Summary And Contributions:**

The paper introduces AVERITEC - a new real-world fact-checking dataset consisting of 4,568 claims, each annotated with question-answer pairs decomposing the fact-checking process, as well as justifications. A multi-step annotation process is used, which guarantees high-quality annotations, providing evidence sufficiency and avoiding temporal leakage. A baseline and an evaluation scheme, are proposed, which launches AVERITEC as a new benchmark.

---

> ### Author Response · Authors · 2023-08-17
>
> We are very happy to hear the reviewer’s positive comments, and delighted that they find the paper well-written and the dataset suitable as a new high-quality benchmark.
>
> Thanks for the suggestion regarding Figure 2 – we have changed it accordingly.

---

### Official Review · Reviewer_qGDa · 2023-07-21
**Review of AVeriTeC**

**Rating:** 6
**Confidence:** 5
**Correctness:** Yes.
**Clarity:** Yes.

**Strengths:**

1.	Describe the strengths of the submission, considering the significance of the contribution, relevance to the broader research community, quality of the research, and ethical and social implications.
2.	This paper proposed a new dataset aimed at fact-checking tasks， wherein the claims are extracted from fact-checkers to guarantee checkworthiness.
3.	In constructing AVeriTeC, the authors ameliorate three issues afflicting existing datasets with real-world claims: Context Dependence, Evidence Insufficiency and Temporal Leaks.
4.	In this dataset, the verdict is determined through question-answer pairs, and it also includes justifications explaining how the evidence supports the given label.
5.	The dataset and benchmark will facilitate progress in fact-checking research on (and likely also beyond) the benchmark tasks.


**Additional Feedback:**

The manuscript refers to the "AVeriTeC score" (in Section 7.2 line 301) but does not provide a clear definition or explanation of what this score represents.

**Documentation:**

Yes.

**Ethics:**

No, there are no or only very minor ethics concerns

**Limitations:**

See Opportunities For Improvement.

**Opportunities For Improvement:**

1.	The baseline system decomposes the claim verdict task into question-answer pairs, which is similar to the approach used in the paper titled "Fact-Checking Complex Claims with Program-Guided Reasoning." It would be possible to compare the baseline results for the AVeriTeC dataset.
2.	The authors' decision to employ the Hungarian METEOR score for evidence evaluation raises the question of why they opted for this metric instead of the more commonly used ROUGE score. It would be better to include a short explanation for the evaluation metric section.
3.	The evaluation of retrieval scores, whether for questions only or questions + answers, raises valid concerns about its reliability. Variations in how different annotators question the claim and subtle differences in the prompt formats used to generate the questions can result in divergent responses. Therefore, the evaluation process should carefully consider lexical, syntax, and semantic factors to ensure a more robust and accurate assessment.
4.	In line 279 “Our baseline, while outperforming the no-search model, does not perform well at higher evidence cutoff points.” It would be better if more discussion about this phenomenon.
5.	The dataset's size is comparatively small, where the claims are primarily collected from fact-checking sources, compared to commonly used datasets like FEVER. To enhance its effectiveness, it would be beneficial to expand the dataset to encompass a broader range of domains and topics.


**Relation To Prior Work:**

Yes.

**Summary And Contributions:**

The authors highlight the shortcomings of existing datasets, which include non-realistic synthetic claims and problematic annotation processes. They then propose the AVeriTeC dataset, which is the first to combine real-world claims with realistic evidence obtained from the web, along with justifications for veracity labels. By meticulously constructing and annotating the data, they address three key issues: context dependence, evidence insufficiency, and temporal leaks. A novel aspect of this work is the decomposition of the fact-checking process into question-answer generation, aligning it with the typical workflow of human fact-checkers. The paper provides ample details on data collection and organization. Furthermore, the authors conduct experiments to validate the performance of baseline models on this dataset, effectively showcasing its difficulty.

---

> ### Author Response · Authors · 2023-08-17
>
> We thank the reviewer for their comments. We are glad to hear the reviewer believes the dataset to be well-constructed and challenging.
>
> **Program-Guided Reasoning Baseline**
>
> We agree the approach taken in the mentioned paper is interesting. However, their strategy assumes a predefined evidence set: all relevant evidence comes from Wikipedia. As such, it is not applicable to AVeriTeC, as evidence in our case must be retrieved from the entire web. Wikipedia does appear as an evidence source in AVeriTeC, but it is used for less than 1% of answers. In AVeriTeC, finding the right knowledge source is a key challenge.
>
> **Evaluation Metric**
>
> We have clarified Section 6 in the paper, in which we discuss our evaluation metric. We chose to create a metric based on the Hungarian algorithm as a generalization of best practices from existing fact-checking tasks (i.e., FEVER) to the case where exact match between retrieved and gold evidence is not a feasible evaluation scheme. Our framework allows evaluation with any text-pair metric scoring function $f$ (see equation 1). Choosing exact match would recover the metric used for evaluating evidence retrieval in FEVER. To generalize to approximate matching, we plug in a standard metric such as METEOR. We chose METEOR over ROUGE as it has been shown to correlate better with human judgments.
>
> We agree that fairly evaluating systems that take a different, equally viable, path through the evidence to the answer is a challenge. We note that it is straightforward to extend our metric to allow multiple reference sets – see the added discussion in Section 6. This would allow measuring performance against multiple viable strategies. However, in general we agree, and as mentioned in Section 6 we consider human evaluation of evidence to be the most reliable approach.
>
> AVeriTeC-score is the name we use as a shorthand to refer to veracity score (accuracy) with evidence cutoff point $\lambda = 0.25$.
>
> **Performance at higher evidence cutoff points**
>
> We have expanded on this section in the paper: “Because of the structure of our pipeline -- generate search terms, retrieve and rerank evidence, generate questions to match the reranked evidence -- our baseline struggles to match specific evidence sets. If the retrieved evidence paragraph is very short, e.g. a table cell reading ‘January 24th’, the question generation model often lacks context to generate the right question. Further, the baseline cannot generate questions with highly abstractive answers, only questions that can be answered directly by sentences in the supporting sources.
>
> **Dataset Size**
>
> As we aimed for real-world fact-checking, a key constraining factor was the amount of (suitable) claims available on the ClaimReview API, which contains the largest collection of fact-checks by fact-checking organizations. Writing a fact-check takes a professional journalist hours, or even days – our process of reverse-engineering allows a crowdworker to construct an example in 20 minutes, but requires fact-checked articles as a starting point. AVeriTeC includes *all* examples of supported claims in the two-year span the dataset covers, from 50 of the most prolific fact-checking organizations around the globe. Working with real fact-checks guarantees that our claims are check-worthy, and we consider this to outweigh the smaller scale.

---

### Official Review · Reviewer_6s6p · 2023-07-21
**Rigorously constructed dataset, baselines slightly unclear**

**Rating:** 7
**Confidence:** 3
**Correctness:** The experiments seem largely correct.
**Clarity:** See the concerns about the baselines …

**Strengths:**

- The dataset is really rigorously constructed. Given the sensitivity around fact-checking, I think the paper does a commendable job of following a principled and sound process for curating this data.
- The paper does a really nice job of describing the annotation process. It would be very easy to replicate this work.
- The paper presents a nice description of the tradeoffs involved in metrics, and motivate their choice of task construction and metrics well.

**Additional Feedback:**

If the authors can answer the above questions regarding the baselines, I'll raise my score.

**Documentation:**

Yes.

**Ethics:**

No.

**Limitations:**

The discussion of limitations is well done.

**Opportunities For Improvement:**

Aspects of the baseline are unclear
- I might have missed this, but how do you control during the search phase to prevent the gathering webpages which essentially fact-check the claim you care about? Or is the idea that it isn’t necessarily a problem if this occurs?
- I’m somewhat unclear on why certain models are being used for certain aspects of the pipeline. For instance, why would use BERT for veracity prediction, and not a foundation model/large language model? My intuition would be that those models perform better, and result in a simpler baseline? Similarly, why wouldn’t you use a LLM for justification generation?

Table 3: I’m not sure I understand the differences between the four rows. How do the first three rows relate to the baseline you describe in 7.1?

Prompts: How many in-context samples did you use for the prompts?

Typos
- L33: “issues”

**Relation To Prior Work:**

Yes. The paper does a very good job here.

**Summary And Contributions:**

- This paper introduces a new dataset of ~4.5k real-world fact-checked claims. Each claim contains extensive justifications and question-answer pairs. This contrasts with previous works, which introduced datasets with “artificial” data that lacked real-world representativeness.
- The authors additionally provide a set of baselines and report performance on the benchmark.

---

> ### Author Response · Authors · 2023-08-17
>
> We are happy the reviewer finds our dataset and annotation process to be rigorous, sound, and principled.
>
> **Baseline details**
>
> The baseline relies on the Google Search API for evidence retrieval, which allows users to request only pages published before a certain date. We use the same strategy during annotation and for the baseline to avoid gathering pages that fact-check or otherwise discuss the claims: we request only pages published before the estimated date of the claim.
>
> We appreciate that the previous version of our paper was a little unclear as to why and how we chose specific models for the various stages of the pipeline. For each stage in the pipeline, we experimented with both BLOOM, Vicuna, BART (for generation), and BERT (for classification). We chose the model that showed the best performance.
>
> Our findings as to which model performs the best in the different stages are indeed somewhat surprising. BLOOM outperforms a larger and newer LLM, Vicuna, on question generation, we suspect because of greater diversity in terms of topics covered by the asked questions. On veracity classification, BERT outperforms Vicuna by a narrow margin (and BLOOM by a larger margin); Vicuna is slightly worse on conflicting evidence, but otherwise performs the same as BERT. On justification generation, BART also outperforms Vicuna and BLOOM – here, we qualitatively find that Vicuna generates as good or better justifications than BART, but the automatic evaluation penalizes Vicuna for its excessive verbosity (producing 50% more tokens per example than BART or the gold standard).
>
> We used 10 in-context examples for our prompts, selected with BM25. We experimented with {1,3,5,10}, and picked the best-performing.
>
> To improve the clarity of our discussion of the baseline, we have substantially edited Section 7.1 in the paper.
>
> **Explanation of Table 3**
>
> The four rows in Table 3 correspond to the experiments discussed in Section 7.2. The third row, “AVeriTeC”, is the baseline discussed in 7.1. The first and second row are variations used for testing: “no search” answers every question with the empty string (but is otherwise identical to 7.1), and “gold evidence” uses the gold question-answer pairs instead of the generated ones (but is otherwise identical to 7.1).
>
> The fourth row, “gpt-3.5-turbo”, measures the performance of ChatGPT on the task (generating both questions, answers, verdicts, and justifications; no search involved) – see the final paragraph of Section 7.2.

---

> > ### Comment · Reviewer_6s6p · 2023-08-24
> >
> > Thanks for the clarification. That's quite helpful. I think it would go a long way in clarifying experimental results if you could include parts of this discussion in the main body, and potentially a more detailed version in the Appendix when you consider the final version.
> >
> > I've adjusted my score accordingly.

---

> > > ### Author Response · Authors · 2023-08-24
> > >
> > > Thank you for your comment. That's excellent news, and we are happy this explanation is clearer! Based on our discussion, we have substantially edited section 7.1 (see the parts in red), as well as the caption for Table 3 and the last paragraph of 7.2. These changes are already live in the current version of the paper -- we hope this fully addresses your concerns.

---

### Official Review · Reviewer_YZ5D · 2023-07-24
**Cool new comprehensive fact checking dataset**

**Rating:** 7
**Confidence:** 4
**Correctness:** seems correct
**Clarity:** yes

**Strengths:**

useful typology of the relevant factors in fact checking datasets. helps organize that literature.

provides a new dataset that will help move this literature forward.

competent experiments and baselines.





**Additional Feedback:**

Nicely done

**Documentation:**

Yes

**Limitations:**

Limitations of evidence based fact checking. what is a claim, etc.

**Opportunities For Improvement:**

could you generate paraphrased data points with GPT?

**Relation To Prior Work:**

Yes

**Summary And Contributions:**

A nice paper that synthesizes and distinguishes different features of fact checking datasets. and then provides a new one that is more comprehensive. They provide a baseline on it.

---

> ### Author Response · Authors · 2023-08-17
>
> We are glad to hear the reviewer finds our dataset competently constructed, useful, and comprehensive.
>
> **Additional ChatGPT-Generated Data**
>
> We thank the reviewer for this suggestion. We agree that generated data is an interesting direction to expand datasets. We conducted an extra experiment, attempting to improve the veracity prediction component by training on paraphrased claims. We kept the evidence and verdict constant, paraphrasing only the claim. Unfortunately, while this did result in increased performance on conflicting evidence, performance dropped for refuted and not enough evidence as the new model often wrongly predicts the former in place of the latter. Overall performance as such dropped by a slight amount. Details of this experiment have been added to Section 7.2 in the revised version.
>
> As the Google Search API is time-consuming and costly to use (three days and $150 for the 1500 claims in the dev and test sets), we did not try generating additional claims to retrieve evidence for. However, we do think this is an interesting direction for future work.

---

### Author Response · Authors · 2023-08-17
**New revision available**

We thank the reviewers for their comments and feedback. We have revised the paper accordingly, and have uploaded a new version. Changes and new paragraphs are highlighted in red.

---

### Decision · Program_Chairs · 2023-09-22

**Decision:**

Accept (Poster)

**Comment:**

Reviewers agree to accept this paper.